# EPiC: Efficient Video Camera Control Learning with Precise Anchor-Video Guidance

## Abstract

Controlling camera motion in video diffusion models is highly sought after for content creation, yet remains a significant challenge. Recent approaches often create anchor videos (i.e., rendered videos that approximate desired camera motions) to guide diffusion models as a structured prior, by rendering from estimated point clouds following camera trajectories. However, errors in point cloud and camera trajectory estimation often lead to inaccurate anchor videos during training. Furthermore, these inherent errors lead to higher training cost and inefficiency, since the model is forced to compensate for rendering misalignments. To address these limitations, we introduce EPiC, an efficient and precise camera control learning framework that constructs well-aligned training anchor videos without the need for camera pose or point cloud estimation. Concretely, we create highly precise anchor videos by masking source videos based on first-frame visibility. This approach ensures strong alignment, eliminates the need for camera/point cloud estimation, and thus can be readily applied to any in-the-wild video to generate image-to-video (I2V) training pairs. Furthermore, we introduce Anchor-ControlNet, a lightweight conditioning module that integrates anchor video guidance in visible regions to pretrained video diffusion models, with less than 1% of backbone model parameters. By combining the proposed anchor video data and ControlNet module, EPiC achieves efficient training with substantially fewer parameters, training steps, and less data, without requiring modifications to the diffusion model backbone. Although being trained on masking-based anchor videos, our method generalizes robustly to anchor videos made with point clouds at test time, enabling precise 3D-informed camera control. EPiC achieves state-of-the-art performance on RealEstate10K and MiraData for I2V camera control task, demonstrating precise and robust camera control ability both quantitatively and qualitatively. Notably, EPiC also exhibits strong zero-shot generalization to video-to-video (V2V) scenarios. This is compelling as it is trained exclusively on I2V data, where anchor videos are derived with only source videos' first frame as visibility referencing. Code is uploaded as supplementary materials. Supplementary videos in https://epic-iclr-submission.netlify.app/.

## 1 Introduction

Recent advancements in video diffusion models (VDMs) (Bar-Tal et al., 2024; Girdhar et al., 2023; Hong et al., 2022; Khachatryan et al., 2023; Wang et al., 2023; Zhang et al., 2024b; Blattmann et al., 2023; Kondratyuk et al., 2023) have significantly improved the generation of realistic videos. As video generation becomes more practical, controlling the process has become a crucial requirement. A key research focus is controlling camera trajectories (Bai et al., 2025a; Yu et al., 2025a; Ren et al., 2025; Shi et al., 2024), which is essential for applications like film recapturing and virtual cinematography. Recent approaches (Ren et al., 2025; Yu et al., 2025a; Cao et al., 2025; Zhang et al., 2024a; Yu et al., 2024b) achieve this by using 3D-informed guidance to create an *'anchor video,'* which approximates the desired camera motion to guide the diffusion model. This method faces challenges, however, as it requires high-quality 3D data from expensive motion-capture systems or relies on inaccurate 3D point cloud/camera trajectory estimators (Wang et al., 2024c; Yang et al., 2024a; Schönberger et al., 2016). These inaccuracies result in pixel-level misalignments between anchor and source videos, which in turn cause training difficulties and inefficiencies (Yu et al., 2025a; 2024b), often requiring extensive computational resources and substantial backbone modifications.

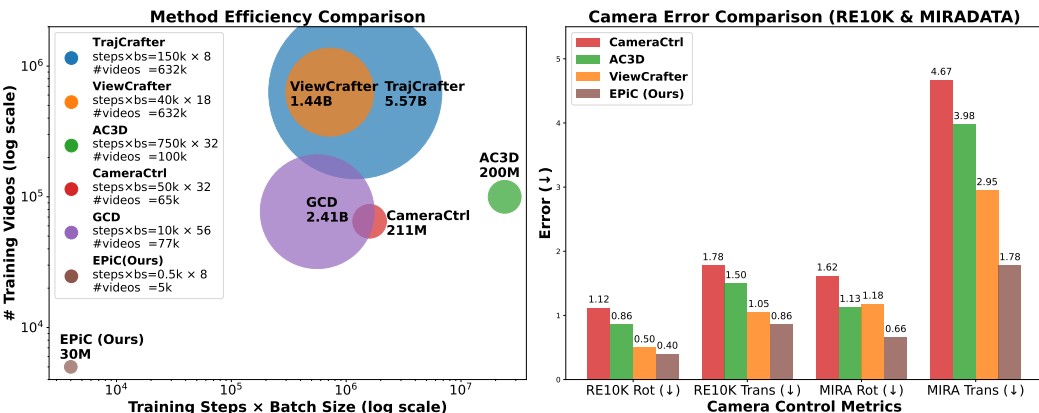

Figure 1: Left: Method efficiency comparison. The circle area is proportional to the number of trainable parameters (exact values are shown below method names). Our method achieves over an order of magnitude higher efficiency in terms of training data, compute cost (steps × batch size), and parameter count. Right: Camera control performance comparison. On both RealEstate10K and Mira datasets, our method achieves the best results with the lowest rotation and transition errors.

Furthermore, most training data mainly comes from multi-view datasets of static scenes (Zhou et al., 2018a; Ling et al., 2024) to ensure high-quality estimations, limiting the models' ability to generalize to real-world videos with dynamic objects (Rockwell et al., 2025).

To address these issues, we propose EPiC, for learning **E**fficient and **P**recise V**i**deo **C**amera control by crafting precisely-aligned training anchor videos with a lightweight, region-aware ControlNet model design (Sec. 4). Our key insight is that anchor videos should be well-aligned with the source videos to make learning as easy, transforming the task from one of more difficult repairing misaligned content to the simpler task of copying visible regions. Thus, unlike previous approaches that render anchor videos from inaccurate 3D point clouds which often misaligned with the source video and reliant on camera trajectories we directly synthesize anchor videos by masking the source video based on first-frame visibility. Specifically, for each subsequent frame, we estimate its pixel trajectories with respect to the first frame from dense optical flow (Teed & Deng, 2020), preserving only those pixels that can be reliably traced back to the first frame. Pixels with no valid correspondence in the first frame are masked out. This process effectively mimics the key property of anchor videos—all new regions relative to the first frame are invisible—while ensuring precise alignment in visible regions. Furthermore, our approach eliminates the need for camera trajectory estimations, allowing anchor videos to be created from any in-the-wild source.

Furthermore, we introduce Anchor-ControlNet (Sec. 4.2), a method that injects anchor-video-based control signals into the generation process with the base model frozen, unlike previous anchor-video-based methods(ViewCrafter (Yu et al., 2024b), Gen3C (Ren et al., 2025) and TrajectoryCrafter (Yu et al., 2025a)) that require extensive full fine-tuning of the backbone. Anchor-ControlNet is a lightweight module with only 26M parameters (<1% of the backbone), injected into the first 25% of backbone layers and using merely 8% of the hidden dimension, direcltly taking the anchor video as control signals. Importantly, to improve quality in invisible regions, we introduce a novel design that makes Anchor-ControlNet visibility-aware by applying visibility masking to its outputs. Specifically, its output is added to the base model's latent representation only within the visible regions, leaving the unseen areas untouched. This design simplifies the ControlNet's task to copying visible content, while delegating the synthesis of occluded or invisible regions entirely to the base diffusion model. This clear division of responsibility prevents errors in invisible regions from influencing the output video, reducing training difficulty and fully unleashing the base model's generative ability in unseen areas. Moreover, restricting ControlNet to visible regions naturally allows user-controlled regional motion—masks on the anchor video can indicate which regions can be moved—thus supporting both static and dynamic scene generation under the same camera trajectory at test time. Combining all these components, we show camera control can be learned with remarkable efficiency: converging with just 5K in-the-wild videos and 500 training steps (less than 5% of the data and steps of prior methods) (Fig. 1 Left), requiring only 15 GPU hours.

Extensive experiments demonstrate that, despite being over an order of magnitude more efficient, EPiC achieves superior performance in camera accuracy (*e.g.*, RotErr, TransErr; Fig. 1, Right) and motion stability (measured by the standard deviation of generated trajectories across different seeds) on image-to-video (I2V) camera control tasks in both indoor and game environments. Moreover, EPiC exhibits strong generalization to video-to-video (V2V) camera control in a zero-shot setting, even though it is trained solely on I2V data. Ablation study shows the effectiveness of our anchor video method and ControlNet design. Our contributions are as follows:

- A novel anchor video construction pipeline with visibility-based masking that produces well-aligned anchor–source video pairs without required point cloud and camera trajectory estimations, while enabling learning from in-the-wild videos.
- A lightweight Anchor-ControlNet with visibility-aware output masking, allowing efficient and precise anchor-video conditioning, as well as selective regional motion control at test time.
- Strong performance on both I2V and V2V camera control tasks with high efficiency in training, data, and model size compared to previous methods.

## 2 RELATED WORK

**Image/Text-Based Camera Control in VDMs.** Controlling camera trajectories in text-to-video (T2V) generation and I2V generation has recently received increasing attention. A common approach is to inject explicit camera parameters (e.g. plücker Embedding) into VDMs (Wang et al., 2024e; Hou et al., 2024b;a; Bahmani et al., 2024b;a; Sun et al., 2024; He et al., 2025b; Zheng et al., 2024; Xu et al., 2024; Watson et al., 2024; Yu et al., 2025b; Li et al., 2025; Zheng et al., 2024; He et al., 2025a; Zhou et al., 2025; Li et al., 2024) for conditioning. However, such parameter-conditioned models often generate world-inconsistent content due to the lack of explicit 3D guidance, especially in out-of-distribution scenarios. To mitigate this, recent works have shifted toward guiding generation with point-cloud renderings (anchor videos) as conditions to leverage geometric cues for more accurate camera control (Yu et al., 2024b; Popov et al., 2025; Hou et al., 2024a; Ren et al., 2025; Zheng et al., 2025; Seo et al., 2024; Cao et al., 2025; Müller et al., 2024; Liu et al., 2024; Zhang et al., 2024a; 2025; Zhou et al., 2024; Yang et al., 2025; Bernal-Berdun et al., 2025). Alternatively, some methods rely on trajectory tracking and encoding as intermediate guidance (Jin et al., 2025; Feng et al., 2024; Xiao et al., 2024; Gu et al., 2025), but such guidance is generally less direct than anchor video conditions and often results in lower accuracy. Despite these advances, rendered anchor videos are often misaligned due to point-cloud errors, and the reliance on accurate camera estimations restricts training to static datasets. Moreover, prior methods require large-scale data to correct misalignment and increase diversity. To address these issues, we propose a masking-based anchor video construction method for precise alignment without camera annotations, and a visibility-aware ControlNet that conditions on the anchor video both efficiently and effectively.

**Video-Based Camera Control.** V2V camera control redirects camera trajectories in existing videos, with applications in filmmaking, augmented reality, and beyond. Unlike T2V and I2V, it is harder to recover comprehensive 4D information from original videos, and paired ground-truth 4D data are scarce. To overcome this, one line of work applies test-time optimization or fine-tuning on specific scenes (You et al., 2024; Zhang et al., 2024a), reducing data reliance but incurring heavy inference overhead. Another line collects large-scale paired videos from simulators such as Unreal Engine5 (Bai et al., 2025a;b), Kubric (Greff et al., 2022; Van Hoorick et al., 2024), or Animated Objaverse (Deitke et al., 2023; Wu et al., 2025; Gao et al., 2024; Yu et al., 2024a; Wang et al., 2024a), though realism and diversity remain limited. The most related works (Bian et al., 2025; Yu et al., 2025a) leverage structured 3D priors (e.g., anchor videos) for controllable V2V generation, but require extensive backbone tuning on large curated 4D datasets. By contrast, our method trains efficiently with only a small amount of I2V data and minimal backbone modification, while generalizing well to V2V.

## 3 BACKGROUND: VIDEO DIFFUSION MODELS

We build on the framework of latent video diffusion models (VDMs), which generate videos by iteratively denoising latent representations in a compressed space. Given an RGB video $x \in \mathbb{R}^{L \times 3 \times H \times W}$, a pre-trained 3D-VAE is used to encode the video into a latent variable $\mathbf{z} = \mathcal{E}(x) \in \mathbb{R}^{L' \times C \times h \times w}$, where $L$ is the number of input frames and $H \times W$ the frame resolution; and $L'$,

$C$, and $h \times w$ the sequence length, channel count, and spatial resolution of the $z$ respectively. Training diffusion models involves learning the reverse of a forward (noising) process. In the forward process, a clean latent sample $\mathbf{z}_0 \sim p_{\text{data}}(\mathbf{z})$ is gradually corrupted with Gaussian noise $\mathbf{z}_t = \sqrt{\bar{\alpha}_t}\,\mathbf{z}_0 + \sqrt{1 - \bar{\alpha}_t}\,\boldsymbol{\epsilon}, \quad \boldsymbol{\epsilon} \sim \mathcal{N}(0, I)$. At each timestep $t$, the model is trained to predict the noise $\boldsymbol{\epsilon}$ from the noisy latent $\mathbf{z}_t$ conditioned on external signals $c$ (e.g., image or text), by minimizing the denoising objective:

$$\mathcal{L}_{\text{denoise}} = \mathbb{E}_{\mathbf{z}_0, t, \boldsymbol{\epsilon}, c} \left[ \left\| \boldsymbol{\epsilon}_\theta(\mathbf{z}_t, t, c) - \boldsymbol{\epsilon} \right\|_2^2 \right] \tag{1}$$

At inference time, the model progressively denoises from Gaussian noise to the final latent representations $\hat{\mathbf{z}}$, which is decoded by the 3D VAE decoder $\mathcal{D}$ to generate the output video: $\hat{\mathbf{x}} = \mathcal{D}(\hat{\mathbf{z}})$.

**Base Model.** We adopt CogVideoX (Yang et al., 2024b) as our base model, which employs a DiT-style (Peebles & Xie, 2023) transformer backbone with full 3D self-attention to jointly model spatial and temporal dependencies across video frames. Specifically, we use the CogVideoX-5B-I2V variant, which supports both image and text conditions for multimodal control during video generation.

**Guiding VDMs with Anchor Video as a Structured Prior for Camera Control.** Recent methods (Yu et al., 2024b; 2025a; Cao et al., 2025; Zhang et al., 2024a) have leveraged *anchor videos* to enable controllable video generation with explicit camera motion control. Anchor videos are typically rendered given camera trajectories from 3D point clouds constructed by lifting a single RGB image into 3D space (Wang et al., 2024b; Yang et al., 2024a). These anchor videos provide explicit geometry and camera motion signals, serving as a structured prior to guide the video generation to follow the intended camera trajectory. During training, the anchor video is created by lifting the first frame of the source video into 3D and rendering it along the source video's camera trajectory. The model then learns to reconstruct the source video conditioned on the anchor video. During inference, the anchor video is constructed similarly using the input image and a user-specified camera trajectory.

However, existing methods face two major challenges: (1) Anchor videos derived from 3D point cloud estimations are often imprecise, leading to difficulties during training ( Fig. 5 (a)). The model must not only inpaint missing regions but also correct misaligned visible areas, resulting in inefficient learning. (2) Conditioning on anchor videos in the latent space typically requires fine-tuning the base model or injecting dense additional modules, which increases computational overhead and reduces model generalization (Tab. 1). To overcome these limitations, we introduce EPiC, a novel and efficient framework for learning precise camera control with masking-based anchor video and a lightweight Anchor-ControlNet, which we will describe in detail next.

## 4 EPiC: An Efficient Framework for Camera Control Learning

Our key idea is to enable controllable video generation through precise anchor-video guidance. Fig. 2 illustrates the overall architecture of our framework. We first construct precisely aligned anchor and source videos as training input-output pairs with a visibility-based masking strategy (Sec. 4.1). Then, we introduce a lightweight Anchor-ControlNet that learns to reconstruct the source video from the anchor video efficiently (Sec. 4.2). Finally, we describe our training and inference details (Sec. 4.3).

### 4.1 Constructing Precise Anchor Videos from Source Videos via Visibility-Based Masking

We aim to construct anchor videos that are well-aligned with the source videos, making the learning process easier and more efficient. To achieve this, we construct anchor videos through a masking strategy that preserves alignment while mimicking the geometric characteristics of point-cloud-rendered videos. Specifically, our process consists of the following two steps:

**Step 1: Pixel-Level Visibility Tracking and Masking.** We estimate pixel trajectories in the source video using dense optical flow from the first frame (computed via RAFT (Teed & Deng, 2020)) to determine whether each pixel remains visible from the original viewpoint. This pixel tracking simulates how content moves or disappears due to viewpoint shifts or occlusion. We provide a binary visibility mask for each frame based on such tracking information, retaining only regions consistently traced from the original view and masking out the rest. This process effectively mimics the core property of anchor videos, which excludes newly revealed content while ensuring precise alignment

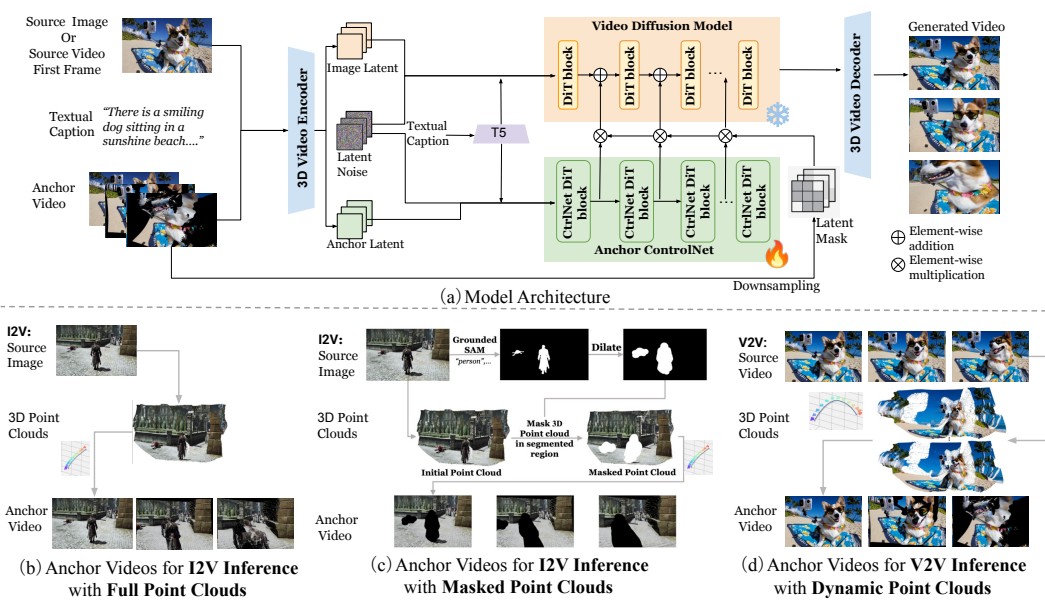

(a) Model Architecture

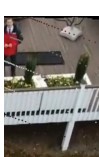

(b) Anchor Videos for **I2V Inference** with **Full Point Clouds**

(c) Anchor Videos for **I2V Inference** with **Masked Point Clouds**

(d) Anchor Videos for **V2V Inference** with **Dynamic Point Clouds**

Figure 2: EPiC Model Architecture. (a) shows an overview of our EPiC framework. EPiC supports multiple inference scenarios. (b) and (c) illustrate our I2V inference scenarios using full and masked point clouds, respectively. (d) depicts V2V inference scenario employing dynamic point clouds.

in the visible regions. In cases where the visible region becomes too small due to large viewpoint shifts, we freeze the mask in subsequent frames to prevent further degradation. The masked source video is obtained by applying the visibility mask to the source video, as shown in Fig. 3.

**Step 2: Artifact Injection.** A major limitation of estimated point clouds is the presence of flying-pixel artifacts, especially around object boundaries (see Fig.2(d), where splatted flying pixels appear near the dog's edges in both point cloud examples). These errors propagate to the anchor video, resulting in flying-pixel artifacts (see Fig.2(d)). To improve robustness, we simulate this flying-pixel effect during training by injecting synthetic dashed rays into the masked anchor video to better align training and inference gap (see Fig. 3 bottom red box). Specifically, we randomly sample a direction and draw multiple rays perpendicular to it, with colors sampled from the first frame to ensure temporal consistency. These

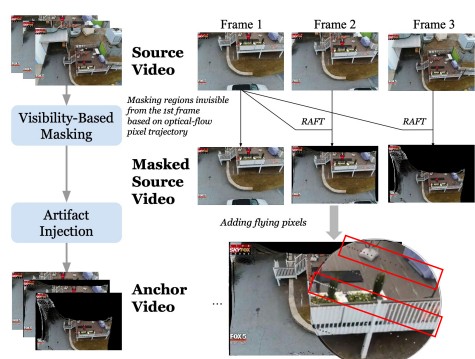

Figure 3: Anchor video construction.

rays are faded and dashed to resemble flying-pixel artifacts, and are applied only within the visible regions defined by the mask, which helps the model learn to ignore such artifacts during inference. The artifact-injected video is used as the final anchor video for training.

## 4.2 GUIDING VIDEO DIFFUSION WITH ANCHOR-CONTROLNET

We introduce Anchor-ControlNet, a variant of ControlNet to guide the base video diffusion model using the constructed anchor video as the condition (Fig. 2 (a)). We follow the principle of using minimal parameters for downstream adaptation to preserve the model's core generation capability (Ruiz et al., 2023) instead of fine-tuning backbone densely. To this end, we adopt a lightweight ControlNet design (<30M parameters) and keep the entire backbone frozen during training.

**Model Architecture.** Anchor-ControlNet is a lightweight DiT-based module designed to inject anchor video guidance into the base diffusion model. Given an anchor video $\mathbf{A}$, we encode it using the 3D VAE from the backbone model to obtain latent features $\mathbf{z}_{anchor}$. During the reverse diffusion process, the noisy latent $\mathbf{z}_t$ is concatenated with $\mathbf{z}_{anchor}$ along the channel dimension. The

combined representation is then patchified and fed into the ControlNet DiT block. The DiT block in Anchor-ControlNet adopts a reduced hidden dimension (256 compared to 3072 in the base model) to maintain efficiency. Its output is projected back to match the backbone's dimension and added to the corresponding layer in the base DiT model. The projection layer is zero-initialized, following the standard practice in ControlNet, to ensure stable integration at the beginning of training.

**Visibility-Aware Output Masking.** Previous work, such as ViewCrafter (Yu et al., 2024b), condition directly on the entire anchor video without visibility awareness. This forces the model to simultaneously repair misaligned regions and inpaint invisible (black) areas, making the learning task unnecessarily difficult and increasing the risk of incorrect region repair during inference(In fact, we also found that simply conditioning on the entire anchor video with ControlNet makes it difficult for the model to learn invisible-region completion, causing it to follow errors present in those invisible areas (Fig. 5 (c))). TrajectoryCrafter (Yu et al., 2025a) incorporates visibility information by encoding the visibility mask into latents, which forces the model to learn the complex relationship among the anchor video, source video, and the mask, thereby increasing training difficulty.

In contrast, with our aligned anchor videos, we can address these issues by manually distinguishing visible and invisible content: the ControlNet focuses solely on copying visible content, while the synthesis of occluded or invisible regions is entirely delegated to the base diffusion model. Formally, we require the control signal from the anchor video to only affect visible regions by applying a binary mask $M \in \{0, 1\}^{T' \times h \times w}$ to the ControlNet output. The mask is downsampled to match the latent resolution and used to selectively update the base model's latent features (Fig. 2a). The ControlNet output is computed as $\tilde{\mathbf{z}} = \text{Proj}(\text{DiT}_{\text{ctrl}}([\mathbf{z}_t, \mathbf{z}_{\text{anchor}}]))$, and then fused with the base model as

$$\hat{\mathbf{z}} = \text{DiT}_{\text{base}}(\mathbf{z}_t) + M \odot \tilde{\mathbf{z}}, \tag{2}$$

where $M$ masks out invisible regions. This visibility-aware latent fusion is applied during both training and inference, allowing the base model to inpaint disoccluded regions while Anchor-ControlNet controls the visible content aligned with the anchor video.

## 4.3 TRAINING AND INFERENCE

In this section, we outline the training and inference paradigm of our framework. EPiC supports multiple inference scenarios, including I2V and V2V, enabling flexible adaptation to diverse applications.

**Training.** We create our masking-based anchor video from in-the-wild source videos to construct training data. We train the Anchor-ControlNet on our collected anchor and source video pairs by conditioning on the anchor video to predict the source video with the training objective in Eq. 1. Details of our in-the-wild video data are provided in Sec. 5.1.

**I2V Inference.** We consider two distinct inference scenarios for I2V: mode (b): **with full point clouds** (illustrated in Fig. 2 (b)) and mode (c) **with masked point clouds** (shown in Fig. 2 (c)). In the first scenario, given an input image and a target camera trajectory, we first estimate the metric depth using DAv2 (Yang et al., 2024a). We then unproject the image into a 3D point cloud and render the anchor video along the specified camera trajectory. However, this approach produces anchor videos where objects remain static, as rendering is performed from a stationary point cloud. For example, the character in Fig. 2 (b) retains the same position and pose throughout the video, limiting its dynamic realism. To overcome this limitation and support **dynamic object movement** while preserving precise camera control, we propose inference with masked point clouds. Specifically, given a single input image, we employ GroundedSAM (Ren et al., 2024) to identify and segment potentially dynamic objects (*e.g.*, "person", "animal") from a predefined category list. Users may also customize tailored segmentation masks. During 3D point cloud projection, we exclude points within the segmented regions (note that we dilate each mask boundary to capture outlier points near the edges). These masked areas are omitted when rendering the anchor video. Our design allows the reserved background to drive camera motion while leaving the segmented foreground objects unconstrained, enabling natural movement within the generated video.

**V2V Inference.** EPiC also supports V2V camera control (Fig. 2 (d)). Given an input video, we apply DepthCrafter (Hu et al., 2024) to estimate continuous depths and construct dynamic point cloud. The anchor video is rendered by replaying the target trajectory over 4D representation. Note that because DepthCrafter predicts depth in each frame's camera coordinate, the reconstructed 4D point cloud is also camera-centric, rather than defined in a global frame. Therefore, the applied trajectory

is interpreted as a relative transformation on top of the source motion. For example, if the original camera moves forward while the control specifies a leftward motion, the resulting trajectory becomes a composition of forward and leftward motions. Additionally, since the base I2V model is frozen, we provide the first frame of the conditional video as input to the model.

# 5 EXPERIMENTS

## 5.1 EXPERIMENTAL SETUP

**Datasets and Baselines.** We compare EPiC and recent baselines for I2V setting on the RealCam-Vid test set (Li et al., 2025) from two data source, RealEstate10K (RE10K) (Zhou et al., 2018b) and MiraData (MIRA) (Ju et al., 2024), consisting of both static and dynamic scenes. We sample 500 videos for each dataset. For baselines, we consider SoTA methods including CameraCtrl (He et al., 2024), AC3D (Bahmani et al., 2024a), ViewCrafter (Yu et al., 2024b), FloVD (Jin et al., 2025), and Gen3C (Ren et al., 2025). For consistency, we use similar anchor videos per test sample for both ViewCrafter and EPiC. For V2V setting, we qualitatively evaluate using Sora videos (Brooks et al., 2024) and challenging movie clips, while provide quantitative results on sampled 100 Kubric4D (Greff et al., 2022) scenes. We use GCD (Van Hoorick et al., 2024), TrajectoryCrafter (Yu et al., 2025a), ReCamMaster (Bai et al., 2025a), and Gen3C (Ren et al., 2025) as V2V baselines.

**Implementation Details.** EPiC is trained on 5,000 videos from the Panda70M dataset (Chen et al., 2024) for 500 iterations, using a total batch size of 16 across 8 40G A100 GPUs. The text condition for the I2V backbone is obtained from the annotated captions in Panda70M. Training takes less than 3 hours with a learning rate of $2 \times 10^{-4}$, using the AdamW (Loshchilov, 2017) optimizer. During inference, we apply classifier-free guidance (CFG) with a scale of 6.0 for text conditioning. More details are in the Appendix Sec. B.1.

**Evaluation Metrics.** For camera-related metrics, we follow prior works (Wang et al., 2024d; He et al., 2024) and report Rotation Error (RotError), Translation Error (TransError), and CamMC, which respectively measure orientation differences, positional errors, and overall camera pose consistency between the predicted and ground-truth trajectories. To account for randomness, we sample five fixed random seeds per test instance and report the mean and standard deviation of each camera metric. For visual quality, we adopt the evaluation protocol from VBench (Huang et al., 2024), including metrics such as Subject Consistency, Background Consistency, Motion Smoothness, Temporal Flickering, Aesthetic Quality, and Imaging Quality. Details of these metrics are provided in the Appendix B.2.

## 5.2 QUANTITATIVE EVALUATION

**Performance.** In Tab. 1, we compare EPiC and recent SOTA I2V camera control methods (CameraCtrl, AC3D, ViewCrafter, FloVD, Gen3C) on RealEstate10K (RE10K) and MiraData (MIRA). EPiC achieves comparable quality scores to those of prior approaches across both the RE10K and MIRA benchmarks. EPiC attains the highest total score on both datasets (82.63 on RE10K and 82.89 on MIRA), suggesting strong subject/background consistency, smooth motion, and reduced temporal flicker. Furthermore, our method significantly outperforms existing baselines in all three camera score metrics. This demonstrates superior fidelity in controlling camera motions, along with the best robustness across seeds, as reflected by the lowest standard deviations.

For V2V camera control, results on Kubric-4D (Tab. 2) show that our method, although only trained on I2V data, is comparable with strong baselines specifically trained for this task such as GCD and TrajCrafter, demonstrating its strong zero-shot generalization ability.

**Efficiency.** In Fig. 1 and Appendix Tab. 4, we present a comparison of training efficiency with the aforementioned methods for I2V and V2V. EPiC requires over an order of magnitude fewer training data and substantially lower training cost, while also using significantly fewer parameters, requiring only 15 GPU hours to train. Importantly, quantitative results show that our method achieves comparable or even superior performance, demonstrating that accurate and robust camera control capability can be achieved without relying on heavy data or computation.

Table 1: Quantitative evaluation results on RealEstate10K (Zhou et al., 2018b) and MiraData (Ju et al., 2024) for I2V camera control. The best numbers are in **bold**. The Total score is the average of all quality metrics. † indicates re-implementation results on I2V.

| Dataset | Method | Total | Quality Score | | | | | | Camera Score | | |
|---|---|---|---|---|---|---|---|---|---|---|---|
| | | | Subject Consist | Bg Consist | Motion Smooth | Temporal Flicker | Aesthetic Quality | Imaging Quality | Rotation Error (↓) | Transition Error (↓) | CamMC (↓) |
| RE10K | CameraCtrl (He et al., 2024) | 78.35 | 89.95 | 91.25 | 97.16 | 91.99 | 43.32 | 56.43 | 1.12 ± 0.44 | 1.78 ± 0.93 | 2.36 ± 1.01 |
| | AC3D† (Bahmani et al., 2024a) | 82.63 | **91.96** | 92.77 | 98.30 | 96.23 | 50.97 | **65.56** | 0.86 ± 0.37 | 1.50 ± 0.82 | 1.97 ± 0.86 |
| | ViewCrafter (Yu et al., 2024b) | 81.18 | 90.23 | 92.99 | 97.74 | 93.51 | 48.29 | 64.33 | 0.50 ± 0.16 | 1.05 ± 0.32 | 1.35 ± 0.40 |
| | FloVD (Jin et al., 2025) | 82.61 | 91.77 | 93.25 | 98.30 | 96.23 | 50.97 | 65.16 | 0.76 ± 0.31 | 1.14 ± 0.52 | 1.47 ± 0.56 |
| | Gen3C (Ren et al., 2025) | 82.27 | 91.10 | 92.75 | 97.99 | **96.67** | 50.61 | 64.54 | 0.45 ± 0.13 | 0.99 ± 0.22 | 1.35 ± 0.30 |
| | EPiC (Ours) | **82.63** | 91.62 | **93.43** | **98.48** | 96.47 | 51.19 | 64.57 | **0.40 ± 0.11** | **0.86 ± 0.18** | **1.17 ± 0.23** |
| MIRA | CameraCtrl (He et al., 2024) | 78.06 | 89.28 | 91.15 | 97.30 | 90.22 | 49.35 | 51.11 | 1.62 ± 0.84 | 4.67 ± 1.47 | 5.66 ± 2.06 |
| | AC3D† (Bahmani et al., 2024a) | 82.78 | 91.75 | 92.81 | 98.20 | 94.77 | 57.64 | **61.51** | 1.13 ± 0.74 | 3.98 ± 1.50 | 4.79 ± 1.53 |
| | ViewCrafter (Yu et al., 2024b) | 79.87 | 86.56 | 91.55 | 96.26 | 91.71 | 54.21 | 58.92 | 1.16 ± 0.34 | 2.95 ± 0.98 | 3.42 ± 1.04 |
| | FloVD (Jin et al., 2025) | 82.55 | 91.64 | 92.91 | 98.43 | 94.67 | 57.46 | 60.21 | 0.95 ± 0.44 | 2.15 ± 0.98 | 3.48 ± 1.03 |
| | Gen3C (Ren et al., 2025) | 80.50 | 88.56 | 90.75 | 96.76 | 91.74 | 55.21 | 59.98 | 0.81 ± 0.24 | 2.05 ± 0.77 | 2.75 ± 0.72 |
| | EPiC (Ours) | **82.89** | 91.82 | 92.94 | **98.75** | 94.86 | **57.94** | 61.03 | **0.66 ± 0.22** | **1.78 ± 0.67** | **2.10 ± 0.60** |

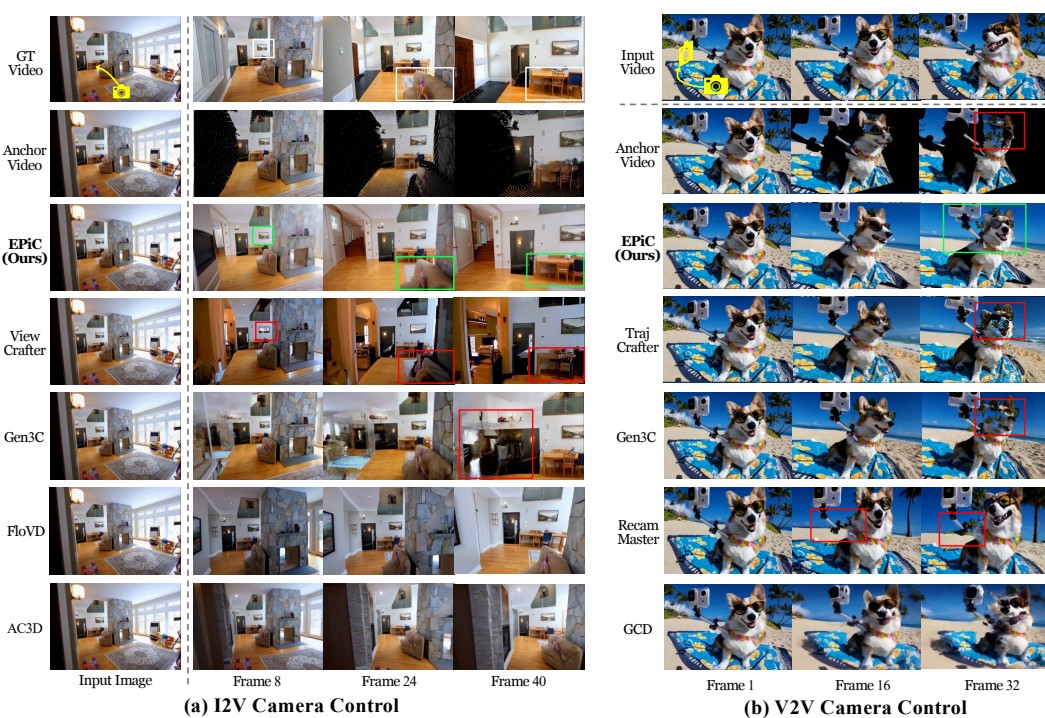

(a) I2V Camera Control    (b) V2V Camera Control

Figure 4: Generated videos comparing with other camera control methods for I2V and V2V tasks.

## 5.3 QUALITATIVE EXAMPLES

Fig. 4 compares camera control results from EPiC and SOTA open-source baselines on both I2V and V2V settings. For I2V, we include ViewCrafter, AC3D, FloVD and Gen3C; for V2V, we compare against GCD, TrajectoryCrafter, Gen3C and ReCamMaster. AC3D, GCD, and ReCamMaster condition on camera embeddings, while ViewCrafter, TrajectoryCrafter, and Gen3C, like ours, condition on anchor videos. FloVD instead uses optical-flow maps as its control signal.

**I2V Camera Control.** As shown in Fig. 4 (a), ViewCrafter (4th row), Gen3C (5th row) our method (3rd row) are capable of following anchor videos. However, as shown in the ViewCrafter row, it often introduces content inconsistencies (red boxes): for example, it gradually changes a painting to glass-like material (2nd column), and produces severe distortions around the sofa (3rd column) and chairs (4th column). Such deviations from the anchor video are potentially due to ViewCrafter learning to over-repair misaligned regions—a side effect of being trained with misaligned point-cloud-based anchor videos. Additionally, Gen3C generates messy content in the invisible region (4th column), struggling with such a large-camera motion scenario. In contrast, our method faithfully preserves visible content thanks to learning from aligned anchor videos (shown in green boxes), and generates reasonable content for the invisible regions. Baseline without anchor video guidance like AC3D and

Table 2: V2V results on Kubric-4D.

| Method | PSNR ↑ | SSIM ↑ |
|---|---|---|
| GCD (Van Hoorick et al., 2024) | 19.72 | 0.59 |
| TrajCrafter (Yu et al., 2025a) | 19.61 | 0.62 |
| EPiC (Ours) | 19.65 | 0.60 |

Table 3: Different anchor video type on Real10K.

| Anchor Video Type | RotErr (↓) | TransErr (↓) | CamMC (↓) |
|---|---|---|---|
| Point cloud-based (1500 iters) | $0.60 \pm 0.20$ | $1.07 \pm 0.39$ | $1.45 \pm 0.62$ |
| Masking-based (500 iters; Ours) | $\mathbf{0.40} \pm 0.11$ | $\mathbf{0.86} \pm 0.18$ | $\mathbf{1.17} \pm 0.23$ |

FloVD, fails to follow the desired camera trajectory. It is worth noting that this example is taken from the RealEstate10K test set, which is an in-domain evaluation setting for ViewCrafter, AC3D and Gen3C, as they are trained densely with RealEstate10K videos. Even so, our method demonstrates superior accuracy and quality. We also provide more qualitative comparisons in Appendix Fig. 14, Fig. 13 and Fig. 12, as well as more in-the-wild examples in Fig. 17.

**V2V Camera Control.** We provide example shown in Fig. 4 (b). While GCD produces blurry foregrounds and lacks fidelity, TrajCrafter, Gen3C, and our method are generally able to follow the anchor video. However, wrong occlusion occurs in the 3rd frame of the anchor video, where the tree passes through regions not reconstructed in the point cloud. Both TrajCrafter and Gen3C incorrectly follow this erroneous signal (red box), potentially due to its heavily modified backbone that enforces anchor-video following even when the renderer is inaccurate. In contrast, our method freezes the entire backbone and only uses the anchor video as guidance, encouraging the model to generate the most plausible content while avoiding being misled by incorrect occlusions (green box). Additionally, ReCamMaster fails to maintain the structure of the selfie stick, while EPiC maintains it successfully thanks to the explicit 3D guidance from the anchor video. We also provide more additional qualitative comparisons and examples on in-the-wild videos in Fig. 15, Fig. 16, Fig. 18, and Fig. 19, as well as single-video multi-camera shooting examples in Fig. 20.

## 5.4 ABLATION STUDIES

In this section, we present ablation studies to validate the key components of our framework. We analyze the impact of different anchor video constructions, artifact injection, visibility-aware output masking, and masked point clouds for dynamic objects. We also provide additional experiments on the effects of training data sources, lightweight model design, generalization to different backbones, and more detailed ablations on Anchor-ControlNet's visibility-aware-output masking in Appendix D.

**Effects of Different Types of Anchor Videos.** We evaluate the effects of different types of anchor videos in Tab. 3 and Fig. 5 (a). For a fair comparison, we select 5K videos with significant camera movement from RealEstate10K, and obtain the anchor video using either a classical point cloud-based method or our visibility-based masking method. We train on point cloud-based anchor videos for 1500 iterations, and masking-based ones for 500 iterations. Tab. 3 shows that training with point cloud-based anchors leads to higher errors and less stable results with larger standard deviation. In Fig. 5(a), due to misalignment, point cloud-based anchor videos lead to slower convergence, producing significantly higher loss than masking-based ones, even with 3× more training. Qualitative results show that models trained with point cloud-based anchors fail to follow the anchor precisely, producing misaligned geometry (red dashed lines in the point cloud-based row), as the model learns an additional task of repairing visible regions, whereas ours faithfully follow (green dashed lines).

**Effects of Artifact Injection for Constructing Training Anchor Videos.** Fig. 5 (b) demonstrates the effectiveness of artifact injection, as described in Sec. 4.1. Due to point cloud estimation errors, flying pixels often appear when rendering from rapidly changing camera poses, resulting in incorrect guidance even within visible regions. Without artifact injection, the model follows these flawed inputs, leading to similar artifacts at inference (red box). In contrast, with artifact injection, the model learns to repair such artifacts during training, resulting in cleaner outputs (green box).

**Effects of Visibility-Aware Output Masking.** One crucial design in our Anchor-ControlNet is the visibility-aware output masking strategy, which enables the model to control only the visible regions, as described in Sec. 4.2. We conduct an ablation study by training modules without mask awareness, similar to ViewCrafter. As shown in Fig. 5 (c), without output masking, the model is influenced by tearing artifacts rendered from the point cloud, which guide it to generate ambiguous content in these corrupted regions (see red boxes). In contrast, our method excludes such regions from the control signal, allowing the model to generate reasonable and faithful content (green boxes).

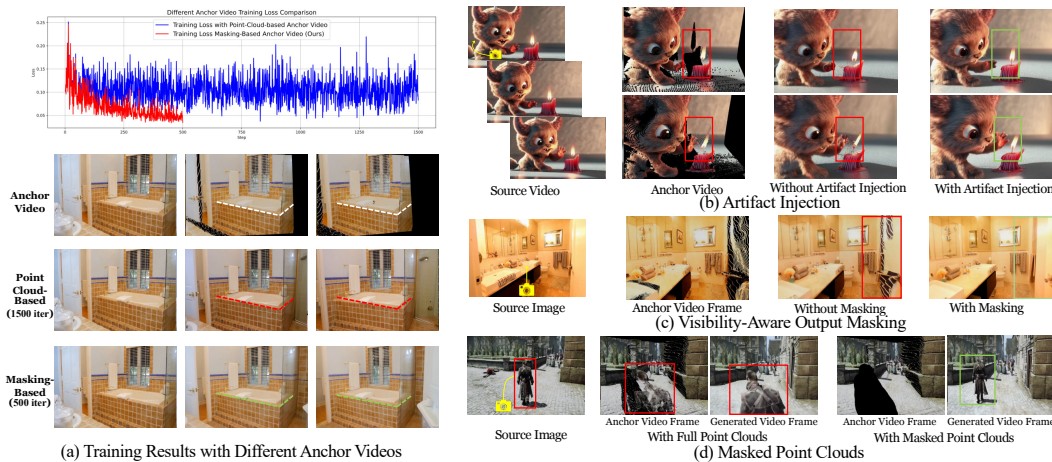

Figure 5: Qualitative examples for ablation study.

**Effects of Masked Point Clouds for Dynamic Objects.** Fig. 5 (d) shows examples of results using the masked point cloud to enable dynamic objects, as described in Sec. 4.3. Without masking (with full point cloud, mode (b) in Fig. 2), the generated video is static—the character (in the red boxes) stands still due to strong 3D guidance in the anchor video. In contrast, masking the point cloud (mode (c) in Fig. 2) removes control signals from the character, allowing it to move freely and enabling a natural walking motion (as shown in the green box). We provide more examples showing our framework's dynamic object control ability in Appendix Fig. 21.

# 6 CONCLUSION

We propose EPiC, an efficient framework for learning camera control. It constructs high-quality training anchors by masking source videos based on first-frame visibility, reducing the need for camera pose estimation and enabling application to in-the-wild videos. We further introduce Anchor-ControlNet, a lightweight adapter that learns to copy visible regions from the anchor video, requiring neither large models, extensive data, nor backbone modifications to correct misalignment. EPiC outperforms previous methods in various visual quality and camera scores. Qualitative experiments in I2V and V2V scenarios, along with comprehensive ablation studies, also validate our design choices.

# 7 ETHICS STATEMENT

This work focuses on efficient and precise camera control in video diffusion models using publicly available or synthetic test datasets such as RealEstate10K, MiraData, Panda70M, and Kubric. No human subjects, personally identifiable information, or sensitive data were involved. All datasets used are released for research purposes and comply with their respective licenses.

The proposed method is designed to improve video generation controllability for applications such as virtual cinematography, content creation, and embodied simulation. While generative models carry potential risks of misuse (e.g., deepfakes or non-consensual content creation), our work primarily targets camera trajectory control, a technical problem that does not inherently amplify these risks. Nonetheless, we encourage responsible use, dataset transparency, and clear labeling of synthetic media. We have no conflicts of interest or sponsorship that could influence the reported results.

# 8 REPRODUCIBILITY STATEMENT

We have made every effort to ensure reproducibility of our results. The training and evaluation datasets (RealEstate10K, MiraData, Panda70M, Kubric) are publicly available. Implementation details, model configurations, and training hyperparameters are fully described in Section 5 and Appendix A of the paper output.

Our method requires only 5K training videos, 500 training iterations, and <20 GPU hours on 8×40GB A100 GPUs, which makes reproduction feasible for most academic labs. Evaluation protocols follow established benchmarks and metrics (e.g., Rotation Error, Translation Error, CamMC, and VBench metrics). Code and supplementary materials (including videos) are provided with the submission to facilitate replication.

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

Figure 6: Comparison between prior 3D point cloud–based anchor video construction and our visibility-based masking approach.

# A  ANCHOR VIDEO CONSTRUCTING METHOD ILLUSTRATION

We provide an illustration of anchor video construction in Figure 6. (a) Previous methods rely on lifting the first frame into a 3D point cloud and rendering along estimated camera trajectories. This often leads to misaligned visible regions due to pose/depth estimation errors, requiring large-scale datasets and many training iterations. (b) In contrast, our visibility-based masking approach directly preserves only pixels that can be traced back to the first frame, producing well-aligned anchor videos without any camera pose estimation. This design greatly simplifies learning and enables efficient training with substantially fewer videos and iterations.

# B  EXPERIMENT DETAILS

## B.1  IMPLEMENTATION DETAILS

EPiC is trained on a subset of $5,000$ videos from the Panda70M dataset (Chen et al., 2024) for 500 iterations, using a total batch size of 16 across 8 40GB A100 GPUs. The text condition for the I2V backbone is obtained from the annotated captions in Panda70M. The subset is selected based on optical flow scores, where we rank videos by their average flow magnitude and retain those with sufficient motion to ensure meaningful camera control training. Training takes less than 3 hours with a learning rate of $2 \times 10^{-4}$, using the AdamW (Loshchilov, 2017) optimizer. For our visibility-aware output masking, we apply average pooling to downsample the raw visibility mask to the latent resolution. We train the Anchor-ControlNet at a resolution of $480 \times 720$ for 49 frames per video (which is the default setting of CogVideoX-5B-I2V (Yang et al., 2024b)), with ControlNet weights set to 1.0.

During inference, we apply classifier-free guidance (CFG) (Ho & Salimans, 2022) with a scale of 6.0 for text conditioning. Following AC3D (Bahmani et al., 2024a), we only inject the ControlNet into the first 40% diffusion steps at inference. We apply max pooling to downsample the raw visibility mask to the latent resolution for visibility-aware output masking. For videos with caption annotations, we directly use the annotations as the textual condition. For those without annotations, we either generate the text condition using advanced vision-language models (Li et al., 2023; Bai et al., 2023) based on the visual input, or manually write prompts for specific usage scenarios.

## B.2  EVALUATION METRICS

We adopt three standard camera pose evaluation metrics to measure the alignment between predicted and ground-truth camera trajectories: **Rotation Error (RotErr)**, **Translation Error (TransErr)**, and **Camera Matrix Consistency (CamMC)** following MotionCtrl (Wang et al., 2024d) and CameraCtrl (He et al., 2024).

Table 4: Efficiency comparison across methods. 'Steps' denotes the number of training iterations, and '#Videos' denotes the amount of training data.

| Method | Steps | Batch Size | Steps×Batch Size | #Videos | #Parameters |
|---|---|---|---|---|---|
| TrajCrafter (Yu et al., 2025a) | 150k | 8 | 1200k | 632k | 5.57B |
| ViewCrafter (Yu et al., 2024b) | 40k | 18 | 720k | 632k | 1.44B |
| AC3D (Bahmani et al., 2024a) | 750k | 32 | 24000k | 100k | 200M |
| CameraCtrl (He et al., 2025a) | 50k | 32 | 1600k | 65k | 211M |
| GCD (Van Hoorick et al., 2024) | 10k | 56 | 560k | 77k | 2.41B |
| Gen3C (Ren et al., 2025) | 10k | 64 | 640k | 100k | 7.23B |
| FloVD (Jin et al., 2025) | 50k | 16 | 800k | 600k | 1.40B |
| ReCamMaster (Bai et al., 2025a) | 20k | 8 | 160k | 136k | 1.49B |
| EPiC (Ours) | 0.5k | 8 | **4k** | **5k** | **26M** |

- **Rotation Error (RotErr)** measures the angular deviation (in radians) between the predicted and ground-truth camera rotations:

$$\text{RotErr} = \sum_{i=1}^{n} \arccos\left(\frac{\text{tr}(\tilde{R}_i R_i^\top) - 1}{2}\right)$$

where $\tilde{R}_i$ and $R_i$ are the predicted and ground-truth rotation matrices at frame $i$, and $n$ is the number of frames in the video.

- **Translation Error (TransErr)** computes the $\mathcal{L}_2$ distance between normalized translation vectors:

$$\text{TransErr} = \sum_{i=1}^{n} \left\| \frac{\tilde{T}_i}{\tilde{s}_i} - \frac{T_i}{s_i} \right\|_2$$

where $\tilde{T}_i$ and $T_i$ are the predicted and ground-truth camera translations, and $\tilde{s}_i$, $s_i$ are their respective scene scales—defined as the $\mathcal{L}_2$ distance between the first and farthest frame in each video.

- **Camera Matrix Consistency (CamMC)** evaluates overall pose alignment by comparing full camera-to-world matrices with scale normalization:

$$\text{CamMC} = \sum_{i=1}^{n} \left\| \left[ \tilde{R}_i \; \frac{\tilde{T}_i}{\tilde{s}_i} \right]^{3\times4} - \left[ R_i \; \frac{T_i}{s_i} \right]^{3\times4} \right\|_2$$

where $\tilde{R}_i$, $\tilde{T}_i$, and $\tilde{s}_i$ are the predicted rotation, translation, and scene scale; $R_i$, $T_i$, and $s_i$ are their ground-truth counterparts.

For visual quality, we adopt the evaluation protocol from VBench (Huang et al., 2024), including metrics such as Subject Consistency, Background Consistency, Motion Smoothness, Temporal Flickering, Aesthetic Quality, and Imaging Quality. We refer to VBench (Huang et al., 2024) for more details.

## C  FULL EFFICIENCY COMPARISON

We provide full efficiency comparison in Table 4. As shown, EPiC achieves over an order-of-magnitude improvement in compute cost, training data size, and parameter efficiency.

## D  ADDITIONAL EXPERIMENTS

In this section, we provide additional ablations on the training data, the use of Anchor-ControlNet, and the lightweight ControlNet design.

### D.1  EFFECTS OF TRAINING DATA SOURCES

A key advantage of our method is that it does not rely on camera pose annotations, which enables training on diverse, in-the-wild video datasets beyond multi-view datasets with limited domain coverage. To validate this, we conduct an ablation comparing training on the widely used

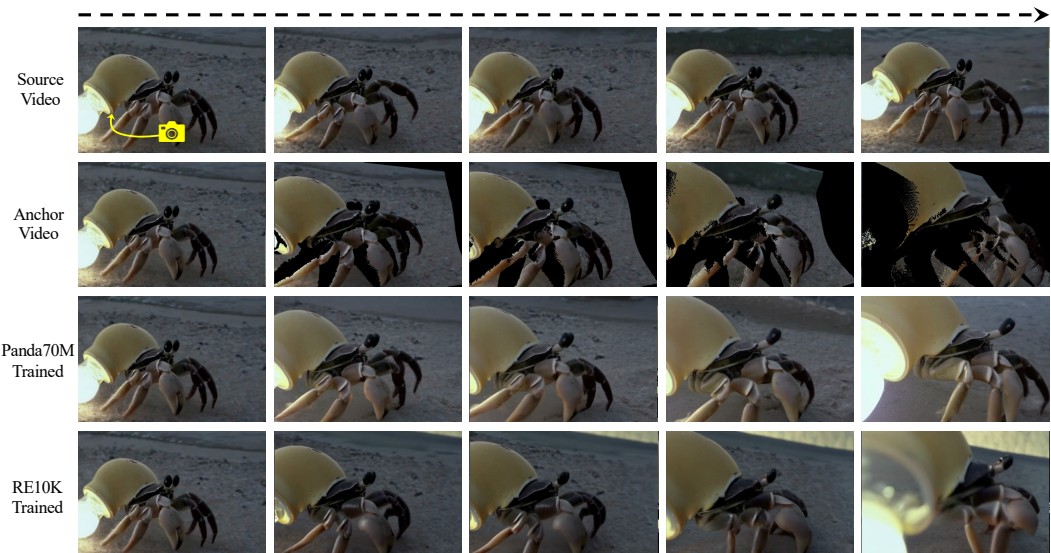

Figure 7: Qualitative V2V camera control results of models trained from different data sources.

Table 5: Ablation of using different data sources for training EPiC.

| Training Data Source | RealEstate10K | | | MiraData | | |
|---|---|---|---|---|---|---|
| | Rot. Err ($\downarrow$) | Trans. Err ($\downarrow$) | CamMC ($\downarrow$) | Rot. Err ($\downarrow$) | Trans. Err ($\downarrow$) | CamMC ($\downarrow$) |
| RealEstate10K Zhou et al. (2018b) | 0.43 ±0.10 | 0.84 ±0.22 | 1.06 ±0.25 | 0.73 ±0.32 | 1.88 ±0.75 | 2.21 ±0.65 |
| Panda70M Chen et al. (2024) | 0.40 ±0.11 | 0.86 ±0.18 | 1.17 ±0.23 | 0.66 ±0.22 | 1.78 ±0.67 | 2.10 ±0.60 |

RealEstate10K (Zhou et al., 2018b), which is a mulit-view dataset limited to static indoor scenes, with training on Panda70M (Chen et al., 2024), which contains more diverse and dynamic videos.

We report quantitative results in Tab. 5. We observe that both data sources yield comparable performance on RealEstate10K, while training with Panda70M achieves slightly better results on MiraData, likely due to its more diverse training content. However, in the V2V setting, especially when the reference video involves fine-grained motion (*e.g.*, detailed limb articulation), models trained on RealEstate10K fail to generalize effectively. Specifically, as shown in Fig. 7, the crab's legs exhibit intricate, localized motion patterns. While the model trained on Panda70M is able to precisely follow these details by following the anchor video, the model trained on RealEstate10K can only capture a coarse moving direction, failing to reproduce the fine motion in the crab's legs. This limitation is likely due to the lack of diverse and dynamic videos in the RealEstate10K dataset, which mainly consists of indoor scenes that differ significantly from the domain of the crab video.

### D.2 EFFECTS OF LIGHTWEIGHT ANCHOR-CONTROLNET DESIGN

We ablate the design of our lightweight ControlNet in Tab. 7. Specifically, we compare injecting into half of the backbone layers (21 layers here (CogVideoX-5B-I2V has 42 layers totally), as in the default ControlNet setting) with and without using pretrained weights, and further study the effect of reducing the number of injection layers. Our results show that using a high-dimensional feature space (3072) with pretrained CogVideoX weights performs comparably to using no pretraining and a much smaller dimension (256), suggesting that the region-copying control is relatively easy to learn. In addition, reducing the number of injection layers to 8 does not hurt performance, while further reducing it to only 2 layers results in a noticeable decreased control accuracy. Based on these findings, we adopt the most cost-effective configuration: injecting into 8 layers with a control dimension of 256.

Table 6: Different video backbones results with EPiC on RealEstate10K dataset.

| Method | Total | Subject Consist | Bg Consist | Quality Score Motion Smooth | Temporal Flicker | Aesthetic Quality | Imaging Quality | Camera Score Rotation Error ($\downarrow$) | Transition Error ($\downarrow$) | CamMC ($\downarrow$) |
|---|---|---|---|---|---|---|---|---|---|---|
| EPiC+CogVideoX (5B) | 82.63 | 91.62 | 93.43 | 98.48 | 96.47 | 51.19 | 64.57 | 0.40 $\pm$ 0.11 | 0.86 $\pm$ 0.18 | 1.17 $\pm$ 0.23 |
| EPiC+Wan2.1 (14B) | 84.24 | 92.97 | 93.54 | 98.53 | 97.42 | 55.67 | 67.34 | 0.41 $\pm$ 0.10 | 0.84 $\pm$ 0.20 | 1.15 $\pm$ 0.21 |

Table 7: Ablation on lightweight ControlNet design. Our selected setting is bolded (no pretrain, 256 hidden dimension, 8 layers).

| Pretrained | Hidden Dimension | #Layers | RealEstate10K Rot. Err $\downarrow$ | Trans. Err $\downarrow$ | CamMC $\downarrow$ |
|---|---|---|---|---|---|
| ✓ | 3072 | 21 | 0.42 | 0.83 | 1.19 |

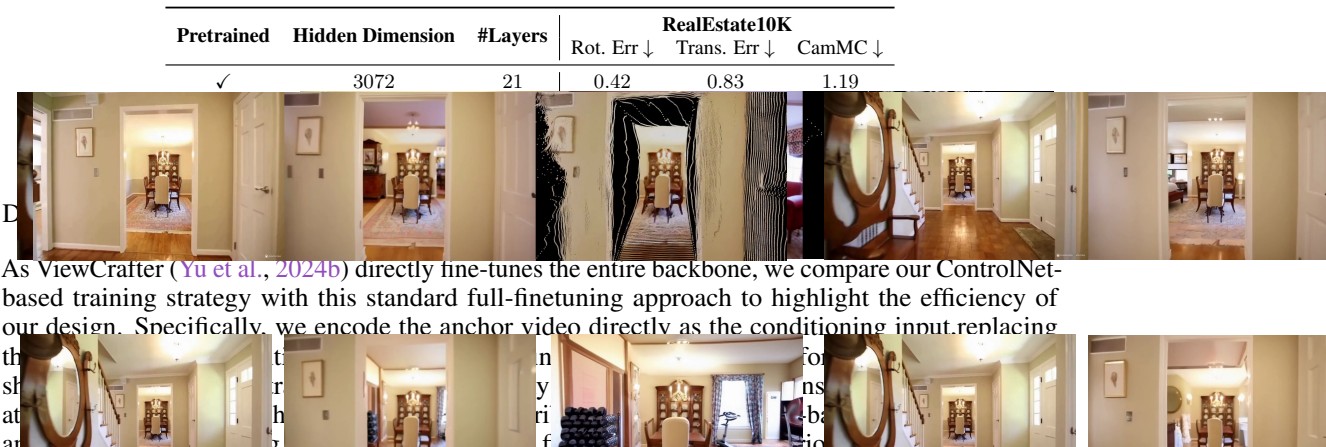

As ViewCrafter (Yu et al., 2024b) directly fine-tunes the entire backbone, we compare our ControlNet-based training strategy with this standard full-finetuning approach to highlight the efficiency of our design. Specifically, we encode the anchor video directly as the conditioning input, replacing

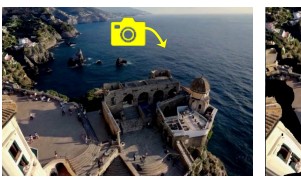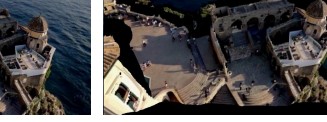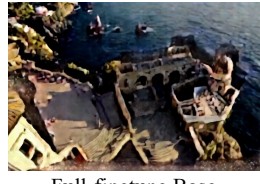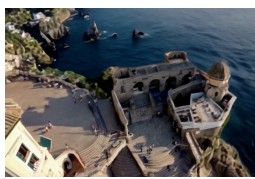

anchor-video conditioning without modifying the backbone, by treating the anchor video as an external control signal.

| Source Frame | Anchor Video Frame | Full-finetune Base Model (1K iter.) | Ours w/ Anchor ControlNet (500 iter.) |

Figure 8: Results of training with Anchor-ControlNet compared to full-finetuning.

## D.4 ADDITIONAL ABLATIONS ON ANCHOR-CONTROLNET'S VISIBILITY-AWARE OUTPUT MASKING DESIGN

We provide further analysis on Anchor-ControlNet's visibility-aware output masking (VAOM) design in Fig. 9. As shown, directly applying a vanilla ControlNet to the anchor video without any masking mechanism causes the model to follow errors in invisible regions, resulting in black or severely white-lined content. This indicates that a plain ControlNet architecture is insufficient for robust anchor-video conditioning. Moreover, applying VAOM only at inference time is also inadequate: it still introduces flickering in several areas, and the invisible regions fail to extend naturally from the visible scene (e.g., in the first example, the black region is completed as a brown patch). In contrast, integrating our VAOM design during both training and inference fully unlocks the base model's ability to complete invisible regions smoothly and coherently, yielding stable, clean, and artifact-free results. This unified training-time integration also enables EPiC to generalize to arbitrary masked anchor videos at test time (Fig. 2), supporting both static and dynamic settings with user-specified dynamic regions.

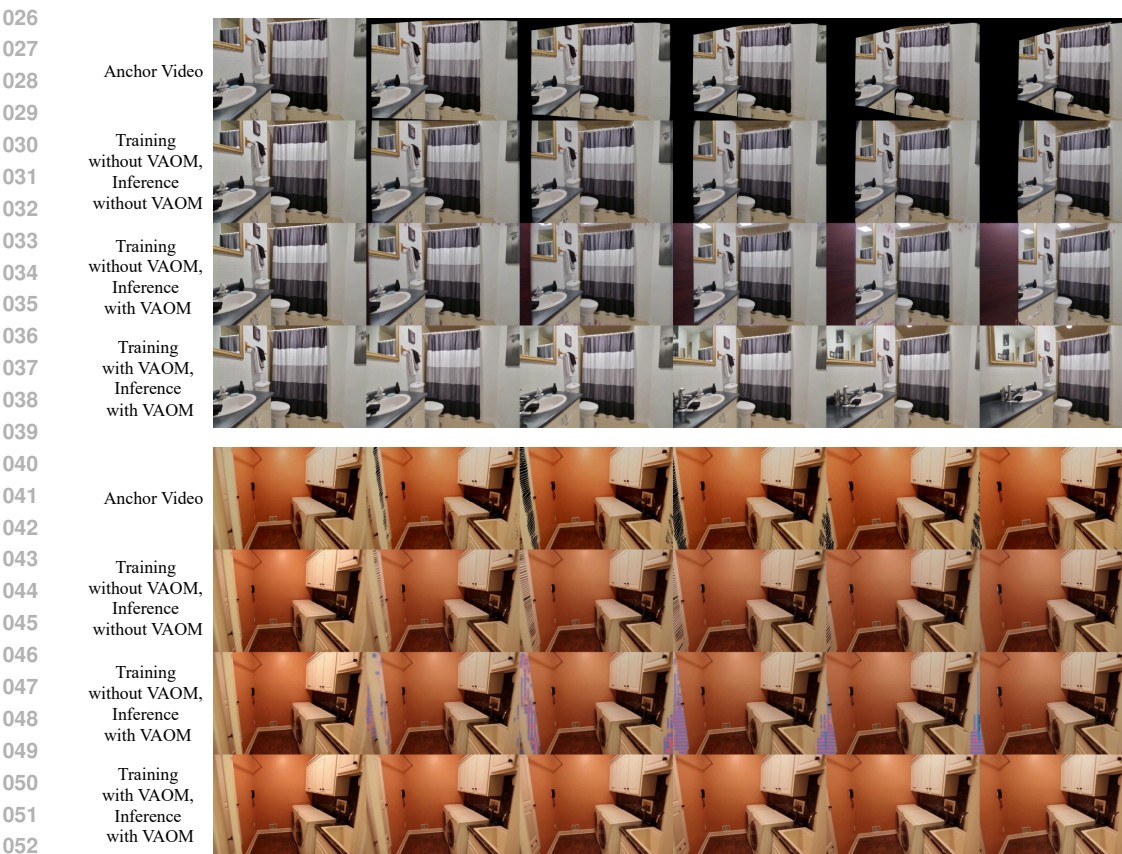

Figure 9: Abalations on Anchor-ControlNet's visibility-aware output masking design.

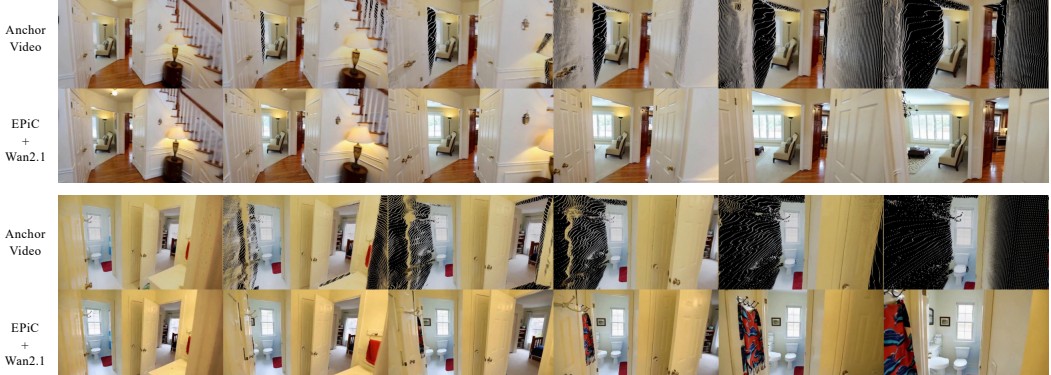

Figure 10: Qualitative results of EPiC with Wan2.1 Backbone on RealEstate10k.

## D.5 GENERALIZATION TO DIFFERENT BACKBONES

We provide additional results to demonstrate EPiC's generalization across different backbones. Specifically, we select Wan-2.1-I2V-14B-480P as the backbone and train EPiC using the same settings. We evaluate the model on the RealEstate10K dataset, and report quantitative results in Tab. 6 and qualitative examples in Fig. 10. As shown, the Wan backbone yields better visual quality while maintaining comparable camera-control accuracy, demonstrating that EPiC generalizes well to stronger base models.

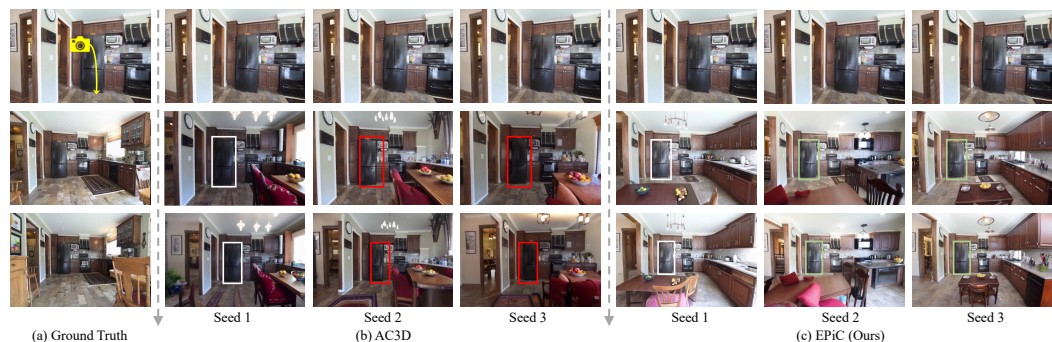

Figure 11: Robustness to different random seeds

# E ROBUSTNESS TO DIFFERENT RANDOM SEEDS

We demonstrate the robustness of our method in Fig. 11. Given a conditioned image, we use a specific object (highlighted with a white box) as the reference for spatial consistency. For AC3D, varying the random seed leads to noticeable changes in the spatial positions of other objects (highlighted in red boxes). This is especially evident in Seed 3, where the generated object's position drifts significantly from the reference, failing to maintain spatial alignment. In contrast, our method consistently preserves the spatial relationship across different seeds. The objects in our generated videos (highlighted in green boxes) remain stable and aligned with the referenced object, demonstrating strong robustness to seed variation.

# F QUALITATIVE COMPARISON WITH BASELINES

## F.1 IMAGE-TO-VIDEO CAMERA CONTROL

**With ViewCrafter.** We provide qualitative comparisons in Fig. 12. While both methods can follow the anchor video, ViewCrafter's visual quality is noticeably lower: in RealEstate10K, it gradually turns a table into a sofa in the first example and makes the toy bear disappear in the second; on MiraData, it often generates messy and unrealistic humans. More examples can be found on our website.

**With FloVD.** We provide qualitative comparisons in Fig. 13. Both EPiC and FloVD share the same CogVideoX-5B-I2V backbone, and their visual quality is generally comparable. However, FloVD struggles to follow the camera trajectory as accurately as ours. We attribute this to its indirect flow-map–based conditioning and the flow-based condition-output misalignment introduced during training. More examples can be found on our website.

**With Gen3C.** We provide qualitative comparisons in Fig. 14. While both methods can follow the anchor video, Gen3C's visual quality is noticeably lower on MiraData. We attribute this to its training data: Gen3C is trained heavily on scene-level datasets, which makes the model behave like a scene-level NVS system and generalize poorly to more dynamic, human-centric content. More examples can be found on our website.

**Controllable Dynamic Objects.** As shown in the examples in Fig. 21, EPiC flexibly supports both dynamic and static scenes in I2V. By contrast, FloVD mainly handles dynamic objects, and Gen3C supports only static scenes. EPiC can naturally do both by simply adjusting the mask in the anchor video to specify which regions should move and which should stay fixed.

## F.2 VIDEO-TO-VIDEO CAMERA CONTROL

**With Gen3C and TrajectoryCrafter.** We provide qualitative comparisons in Fig. 15. In the first example, both Gen3C and TrajectoryCrafter follow the anchor video too rigidly, resulting in a half-body mammoth or incorrect occlusions caused by erroneous anchor-video rendering. We attribute

this to their full-finetuning strategy, which turns the models into strict anchor-following systems with weakened semantic priors. In contrast, EPiC follows the anchor video while still generating semantically coherent content, thanks to its frozen-backbone design that preserves strong first-frame semantic priors. More examples can be found on our website.

**With ReCamMaster.** We provide qualitative comparisons in Fig. 15. We observe several issues with RecamMaster (1) Without explicit 3D guidance, it struggles to maintain correct geometry, as shown in the first example where the selfie stick becomes distorted; (2)As its conditioning is based on absolute camera parameters, it fails on videos with camera motion (second example), causing both the moving camera and the SUV to appear static; (3) it hallucinates objects not present in the source video (third example), such as an extra basketball and even a nonexistent backboard; and (4) it sometimes produces oil-painting-like artifacts (fourth example). In contrast, EPiC generates more natural and stable results without these issues, thanks to the explicit anchor-video guidance and the strongly maintained first-frame semantic prior. More examples can be found on our website.

# G ADDITIONAL QUALITATIVE RESULTS

**I2V Qualitative Examples.** We showcase diverse qualitative examples of I2V camera control spanning a wide variety of scenarios in Fig. 17, including daily-life activities (cooking, dining, exercising), human–animal interactions (fox resting, horse walking), transportation (cycling, subway), outdoor navigation (kayaking, hiking, urban scenes), and complex virtual environments (video games, historical architectures, and futuristic cityscapes). These examples highlight that EPiC can handle both indoor and outdoor scenes, real-world and synthetic data, and static as well as dynamic objects. The results demonstrate strong generalization across highly diverse contexts, producing coherent motion and faithful camera control without overfitting to specific domains. More examples can be found on our website.

**V2V Qualitative Examples.** We present diverse examples of V2V camera control spanning movie clips and in-the-wild videos in Fig. 18 and Fig. 19. Across various camera trajectories, our method is able to faithfully follow the target motion while producing high-quality and visually coherent results. More examples can be found on our website.

**V2V Multi-Camera Shooting.** We further demonstrate multi-camera shooting in Fig. 20, where multiple trajectories are generated from a single input video. The results show strong temporal consistency across different camera views, indicating that our method can maintain coherent scene structure and appearance under diverse camera motions. More examples can be found on our website.

**I2V Inference Modes.** We show results of different I2V inference modes (mode (b) and (c) in Fig. 2) in Fig. 21. With the full point cloud in mode (b), our method tends to generate static content. By masking the point cloud in mode (c), we can make specific objects or background dynamic, demonstrating the ability to control both object motion and scene dynamics. More examples can be found on our website.

**Examples of Constructed Anchor Videos.** We present examples of high-quality anchor videos constructed from Panda70M source videos in Fig. 22. Our method consistently maintains spatial coherence and masks regions that were initially not visible in the first frame, even when objects exhibit significant movements across frames, while the Panda70M provides both diverse and dynamic video data. Such high-quality and diverse anchor videos further help the efficient learning by our model. Video examples can be found on our website.

# H ADDITIONAL APPLICATIONS: FINE-GRAINED CONTROL

We present several additional applications demonstrating different types of fine-grained control based on a single image with our anchor-video conditioning.

**Text-Guided Scene Control.** Our model effectively demonstrates dynamic text-guided video generation capabilities, enabling flexible scene synthesis across different styles while maintaining

temporal and spatial consistency. Fig. 23 illustrates examples of our text-guided scene control. Starting from an initial frame with a fixed forward camera trajectory, our method generates subsequent video frames conditioned on different textual prompts. The newly prompted objects are introduced into the generated scene (highlighted in red text and boxes), while the objects present in the initial frame remain consistently visible throughout the video (highlighted in green text and boxes).

**Object 3D Trajectory Control via Anchor Video Manipulation.** We also demonstrate the flexibility of our method in enabling 3D trajectory control for objects. The input is usually a 3D trajectory (*e.g.*, indicating moving backwards with 2 meters) applied to a specific object (*e.g.* corgi). We encode the desired motion into the anchor video by manipulating it based on the 3D trajectory. Specifically, following a similar approach to our inference setup with masked point clouds, we use GroundedSAM (Ren et al., 2024) to obtain the segmentation mask of the corgi, extract the point cloud corresponding to the corgi, and isolate the background point cloud without the corgi. We then simulate motion by translating the corgi's point cloud backward by 2 meters relative to the background over time (we don't move the background point cloud), producing a dynamic point cloud sequence for rendering. In this setup, we focus solely on trajectory control, thus, we remain the camera trajectory static during rendering. The resulting anchor video depicts the corgi moving backward and serves as strong guidance. Our results are illustrated in Fig. 24, where our approach successfully generates scenarios in which the corgi steps backward. In contrast, AC3D, which conditions only on camera embeddings, which lack explicit trajectory information, fails to generate this backward motion even with "stepping backward" included in the textual condition. This comparison highlights the strength of our method in interpreting and executing precise object-level movements in 3D space, showcasing its superior capability for controllable video generation.

**Regional Animation.** Our method is also applicable to regional image animation, where motion is localized to a specific area based on a short text prompt and a user-provided click or prior mask. To achieve this, we directly create the anchor video by repeating the source image and applying the regional mask to each frame. As shown in Fig.25 (a), given the prompt "the corgi shakes its head," with corresponding corgi head mask, our method generates a video in which only the corgi's head moves while the rest of its body remains still, accurately following both the textual instruction and the specified region. In contrast, Fig.25 (b) highlights a failure case of AC3D—when the intended motion is for the palm tree to move, AC3D incorrectly animates the corgi instead. Our method, however, successfully isolates and animates the palm tree, demonstrating its ability to localize motion precisely based on regional guidance and text. This showcases the fine-grained spatial control ability enabled by our approach.

## I   FAILURE ANALYSIS

Since our model learns to follow the anchor video in visible regions, it can be affected when the estimated point-cloud structure or occlusion masks are inaccurate. We provide two examples in Fig. 26 on the website illustrating the main failure modes: (1) **Incorrect point-cloud structure.** In the first example, a misestimated point cloud causes the man with a backpack in the anchor video to appear tilted, and our result partially inherits this (e.g., a slightly stretched neck). The face of the person next to him also begins to tilt. In comparison, ViewCrafter loses track of the motion and produces randomly distorted humans, while Gen3C strictly follows the erroneous structure, resulting in even more distorted outputs. EPiC, despite inheriting some of the structural bias, remains noticeably more stable. (2) **Incorrect occlusion.** In the second example, background color leaks through the kangaroo's face in the anchor video. EPiC interprets this as a mild blue lighting effect, whereas TrajectoryCrafter and Gen3C rigidly copy the artifact and produce visible holes in the face. These analyses clarify how EPiC behaves under imperfect 3D estimation and demonstrate that—even in failure cases—it remains more robust than baseline methods.

## J   LIMITATIONS AND BROADER IMPACTS

EPiC trains a lightweight adapter on a backbone video diffusion model. As such, its performance, output quality, and potential visual artifacts are inherently influenced by the capabilities and limitations

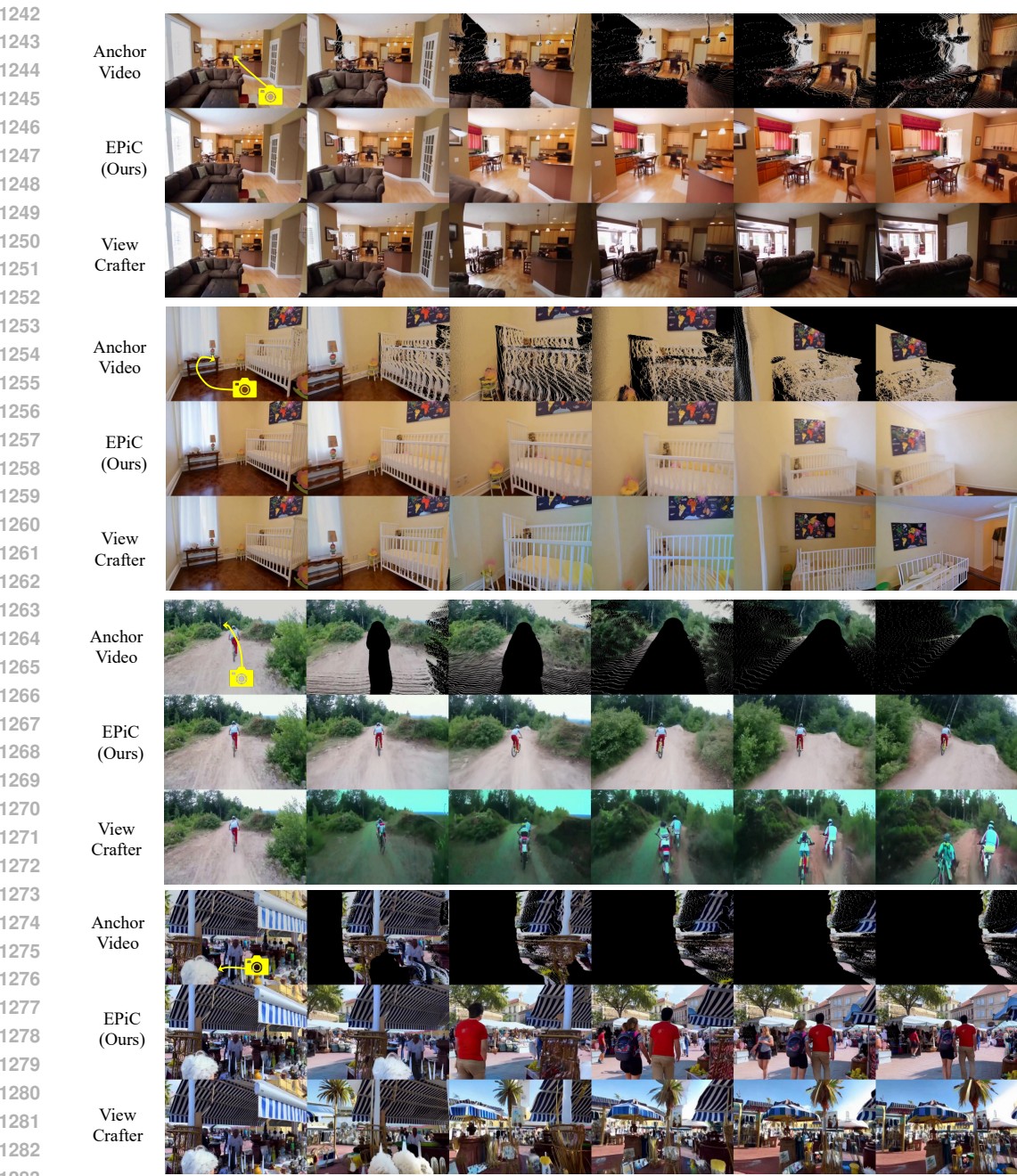

Figure 12: I2V Comparison with ViewCrafter. The first two examples are from RealEstate10K, while the last two examples come from MiRaData.

of the underlying backbone models it relies on. For instance, if the backbone model struggles with generating complex, rare, or previously unseen scenes and objects, then EPiC may also exhibit suboptimal generation results. This dependency highlights the importance of selecting strong and reliable backbone models when applying EPiC.

While EPiC can benefit numerous applications in video generation, similar to other visual generation frameworks, it can also be used for potentially harmful purposes (e.g., creating false information or misleading videos). Therefore, it should be used with caution in real-world applications.

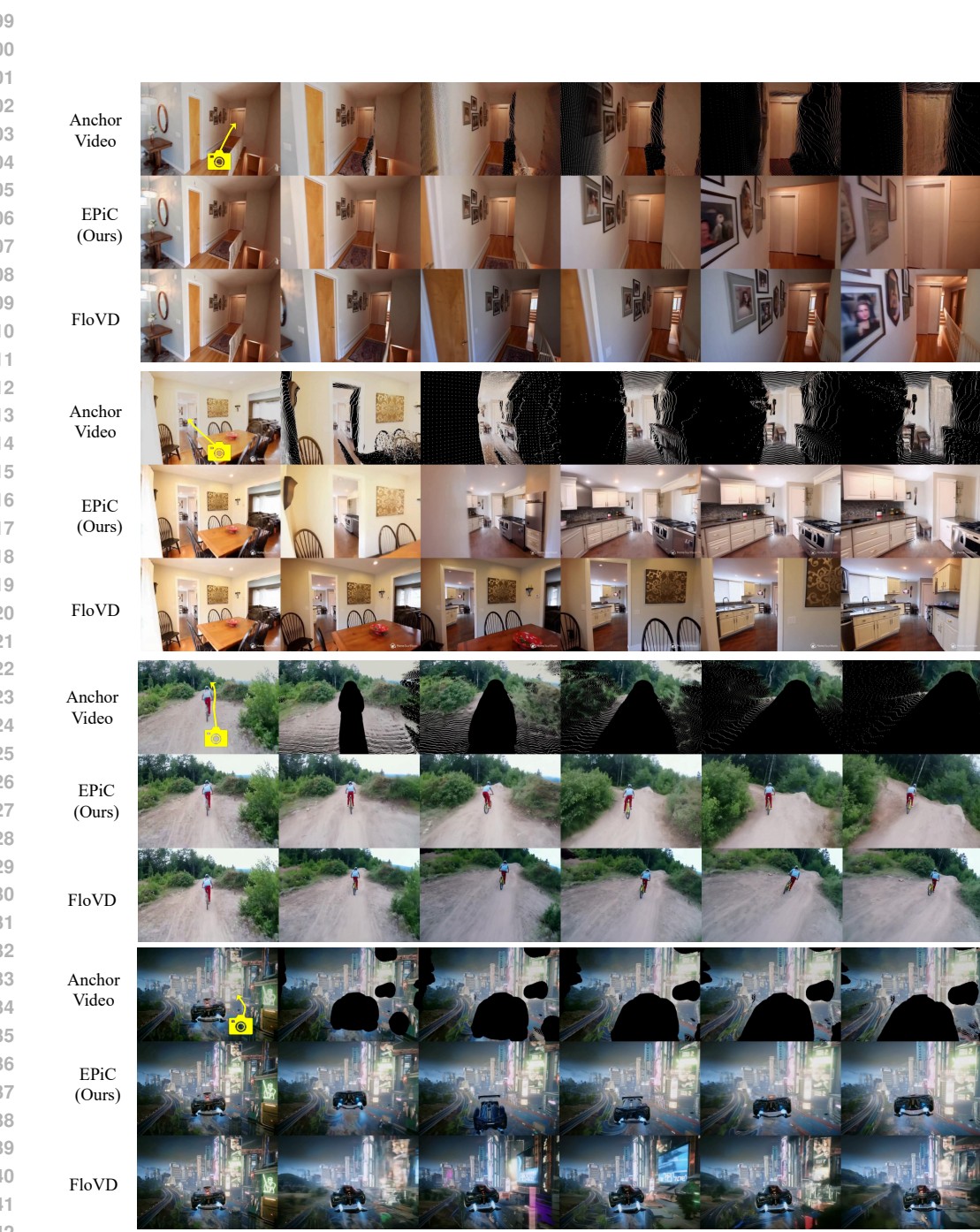

Figure 13: I2V Comparison with FloVD. The first two examples are from RealEstate10K, while the last two examples come from MiRaData.

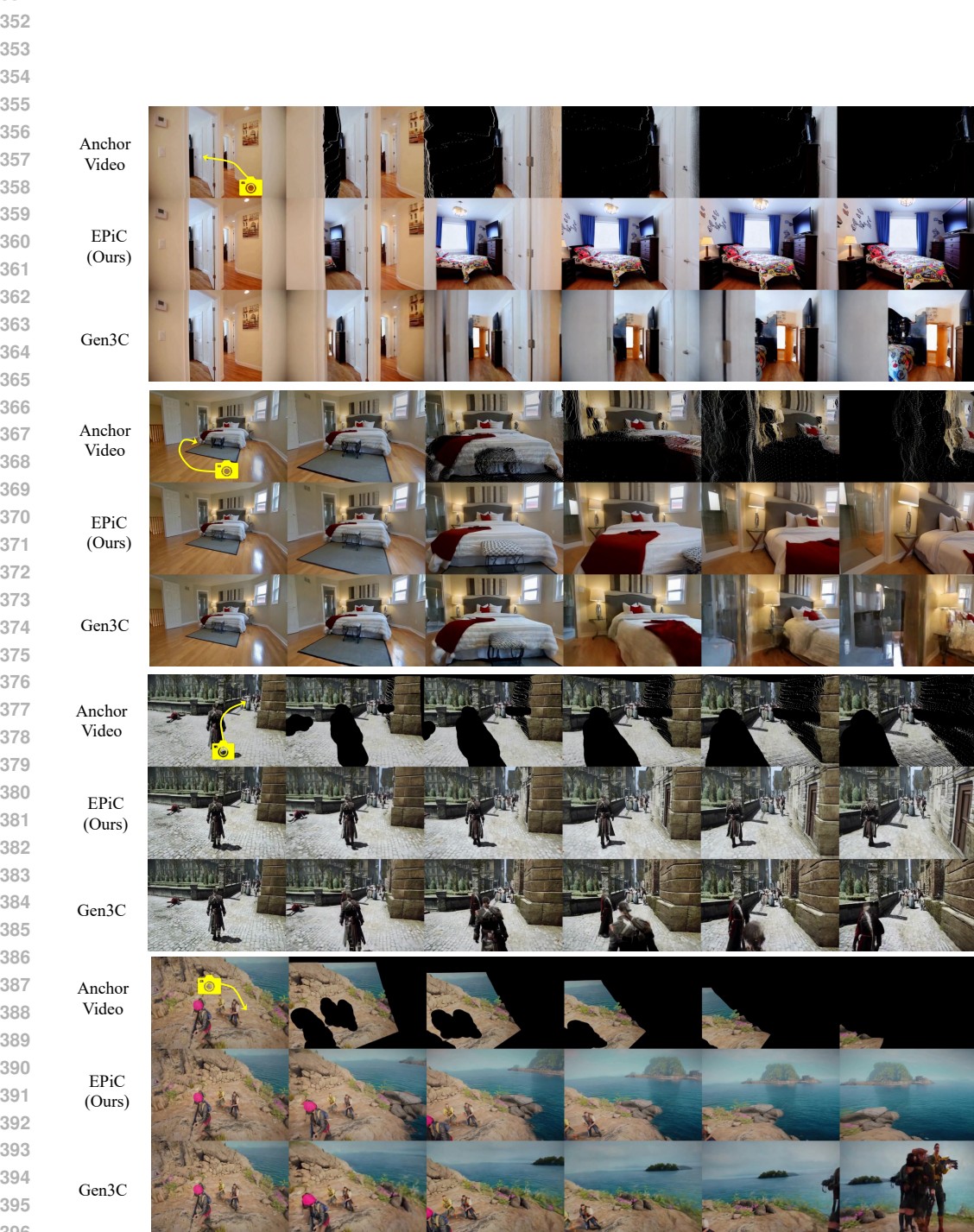

Figure 14: I2V Comparison with Gen3C. The first two examples are from RealEstate10K, while the last two examples come from MiRaData.

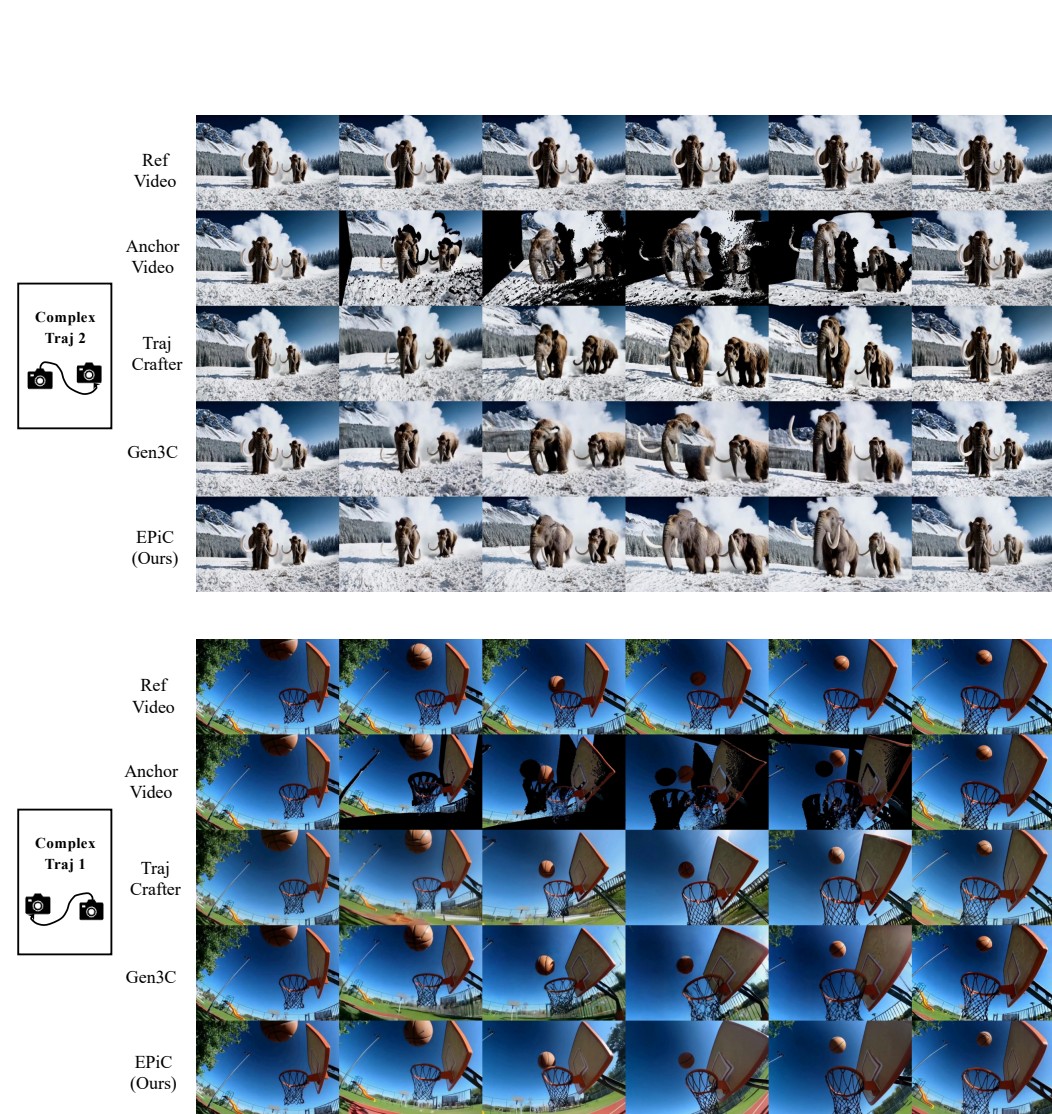

Figure 15: V2V Comparison with Gen3C and TrajectoryCrafter.

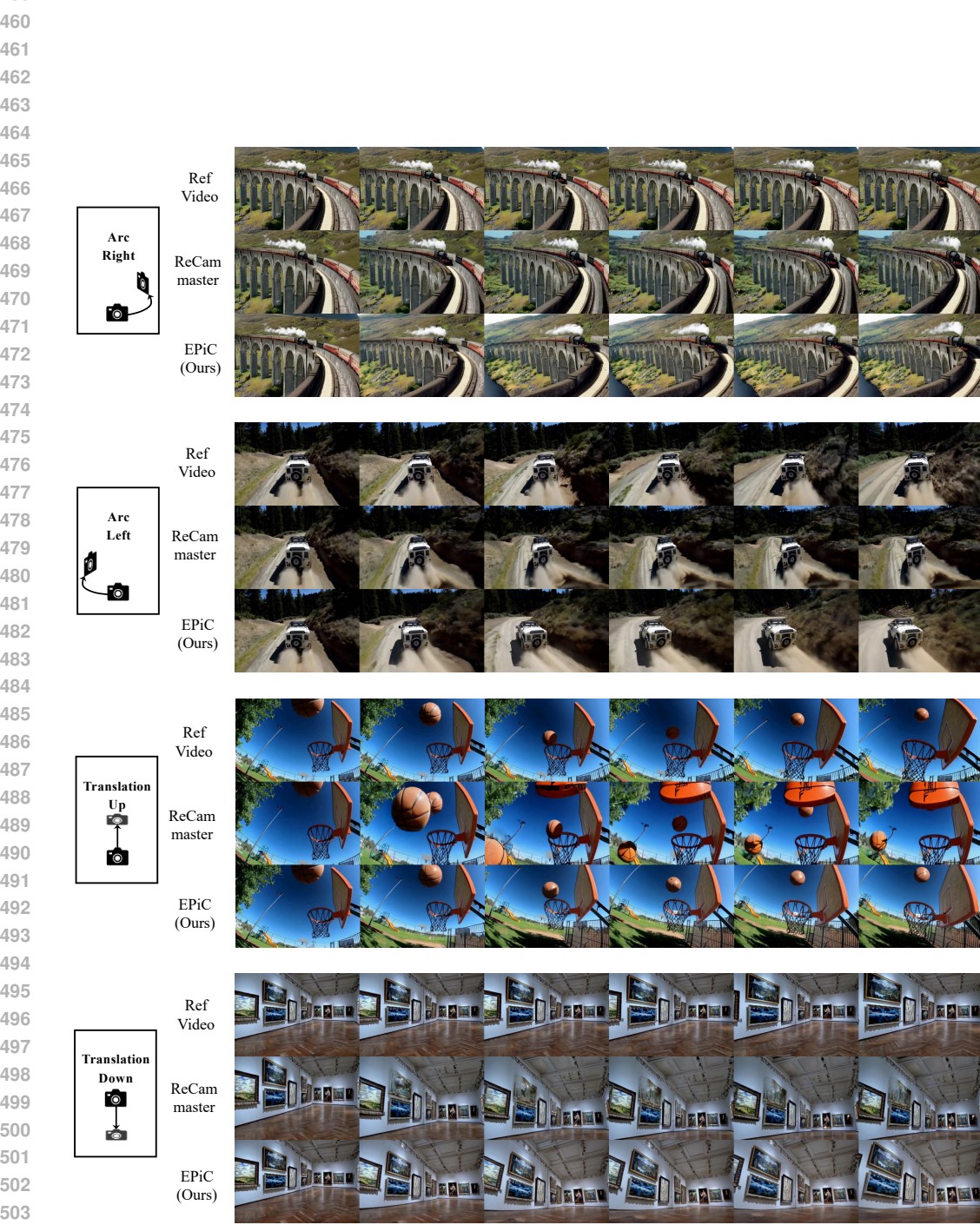

Figure 16: V2V Comparison with ReCamMaster.

1512
1513
1514
1515
1516
1517
1518
1519
1520
1521
1522
1523
1524
1525
1526
1527
1528
1529
1530
1531
1532
1533
1534
1535
1536
1537
1538
1539
1540
1541
1542
1543
1544
1545
1546
1547
1548
1549
1550
1551
1552
1553
1554
1555
1556
1557
1558
1559
1560
1561
1562
1563
1564
1565

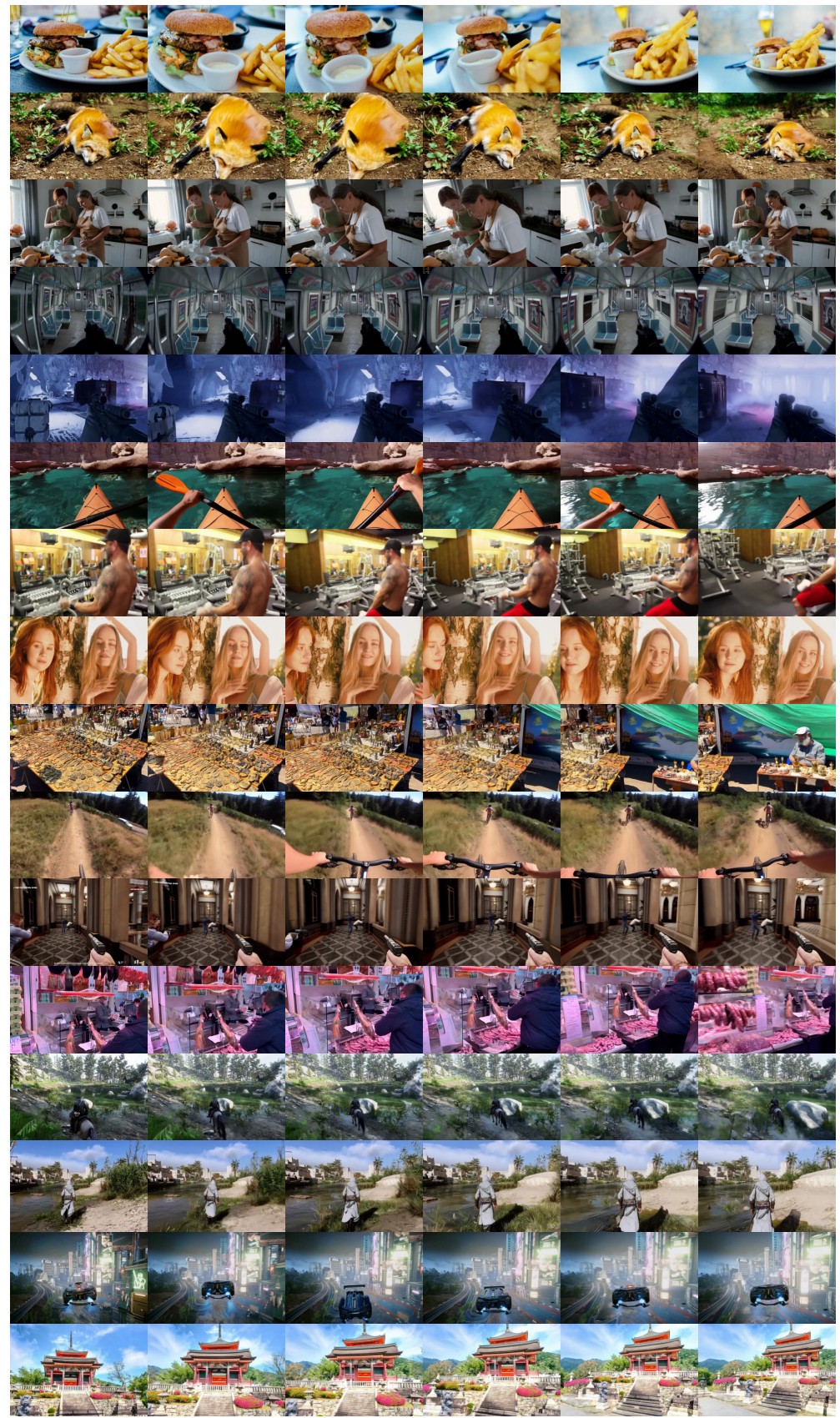

Figure 17: Diverse I2V camera control results.

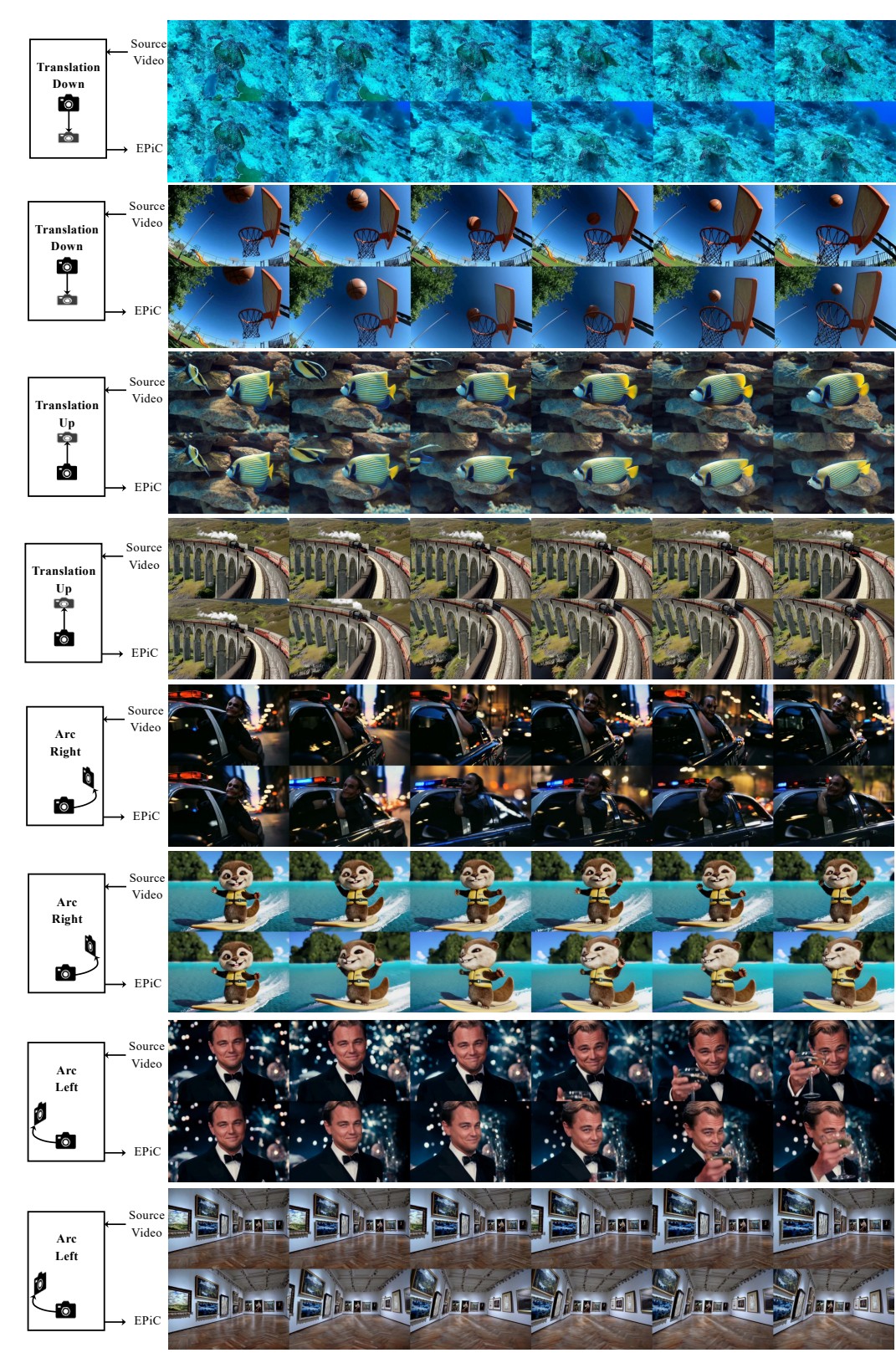

Figure 18: Diverse V2V camera control results.

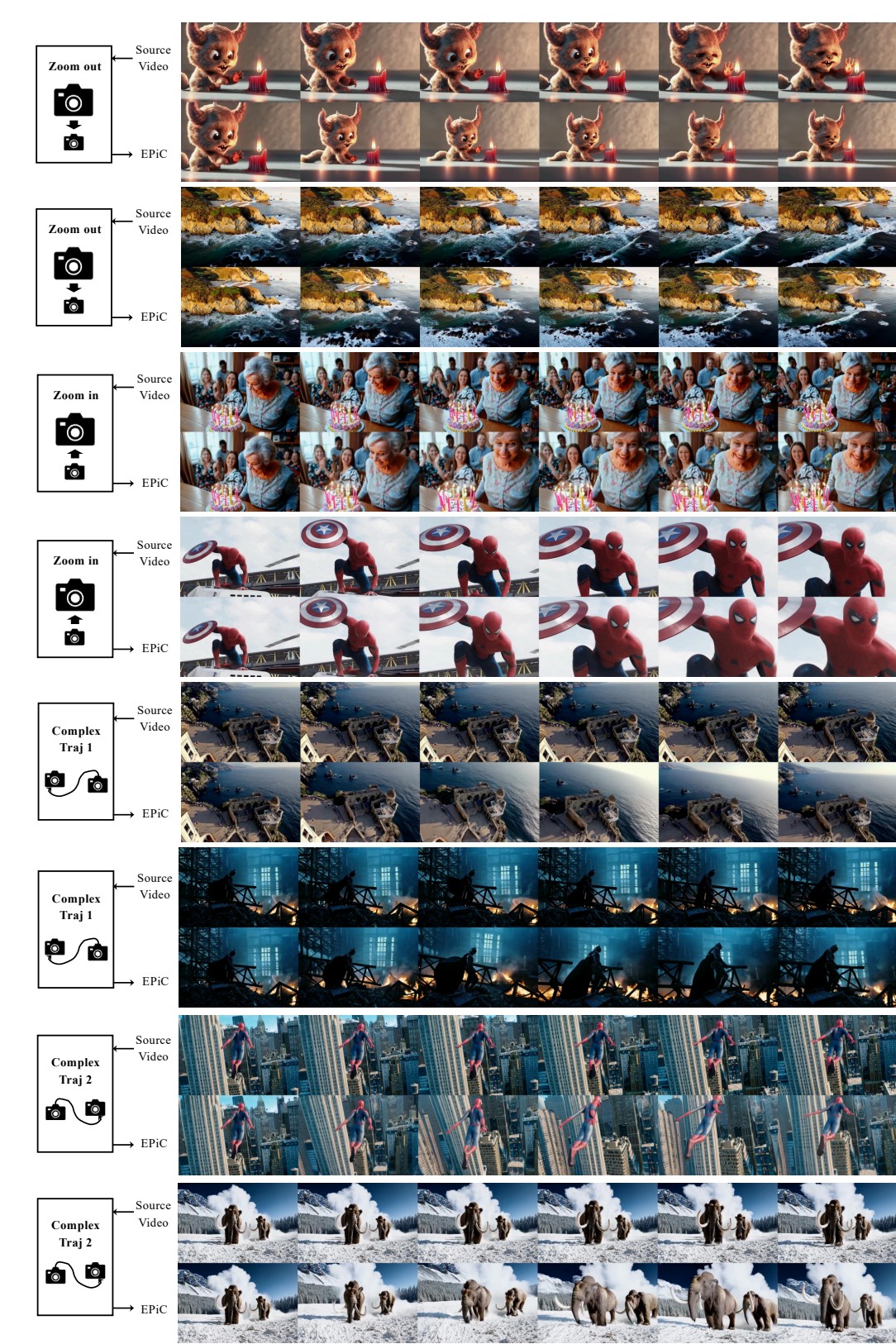

Figure 19: Diverse V2V camera control results.

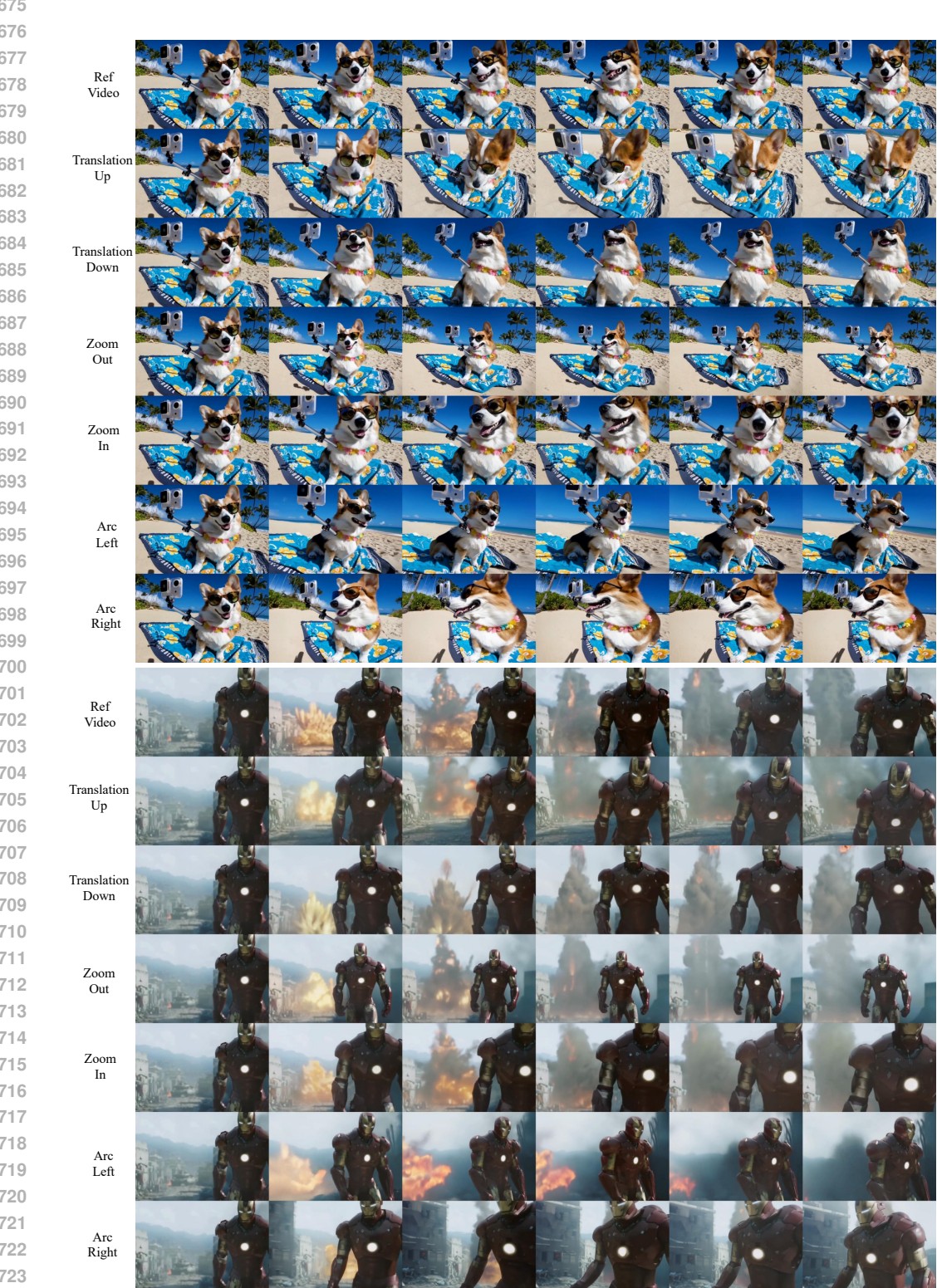

Figure 20: Multi-camera shooting examples for V2V.

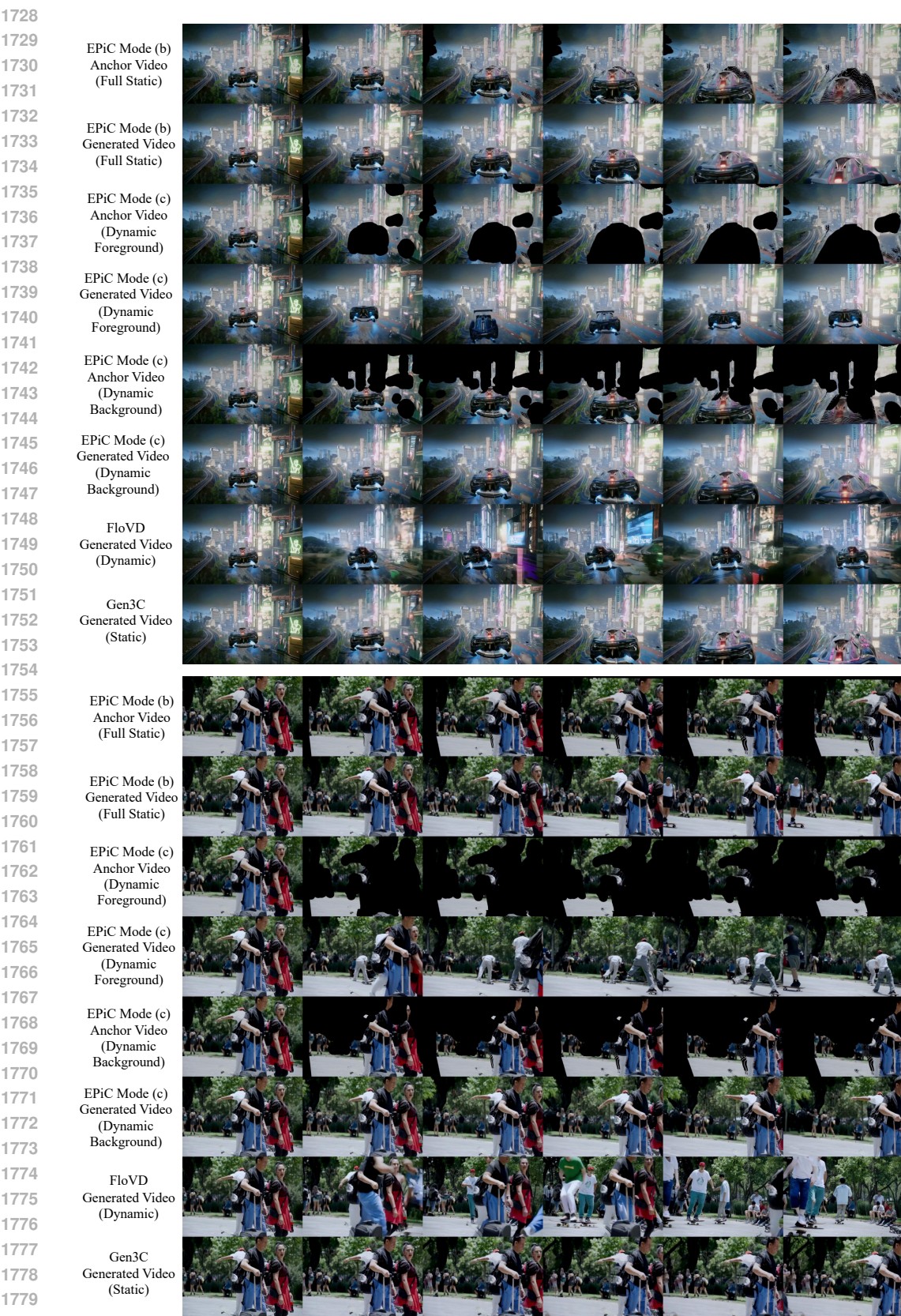

Figure 21: Inference with different I2V modes as well as comparison to baselines.

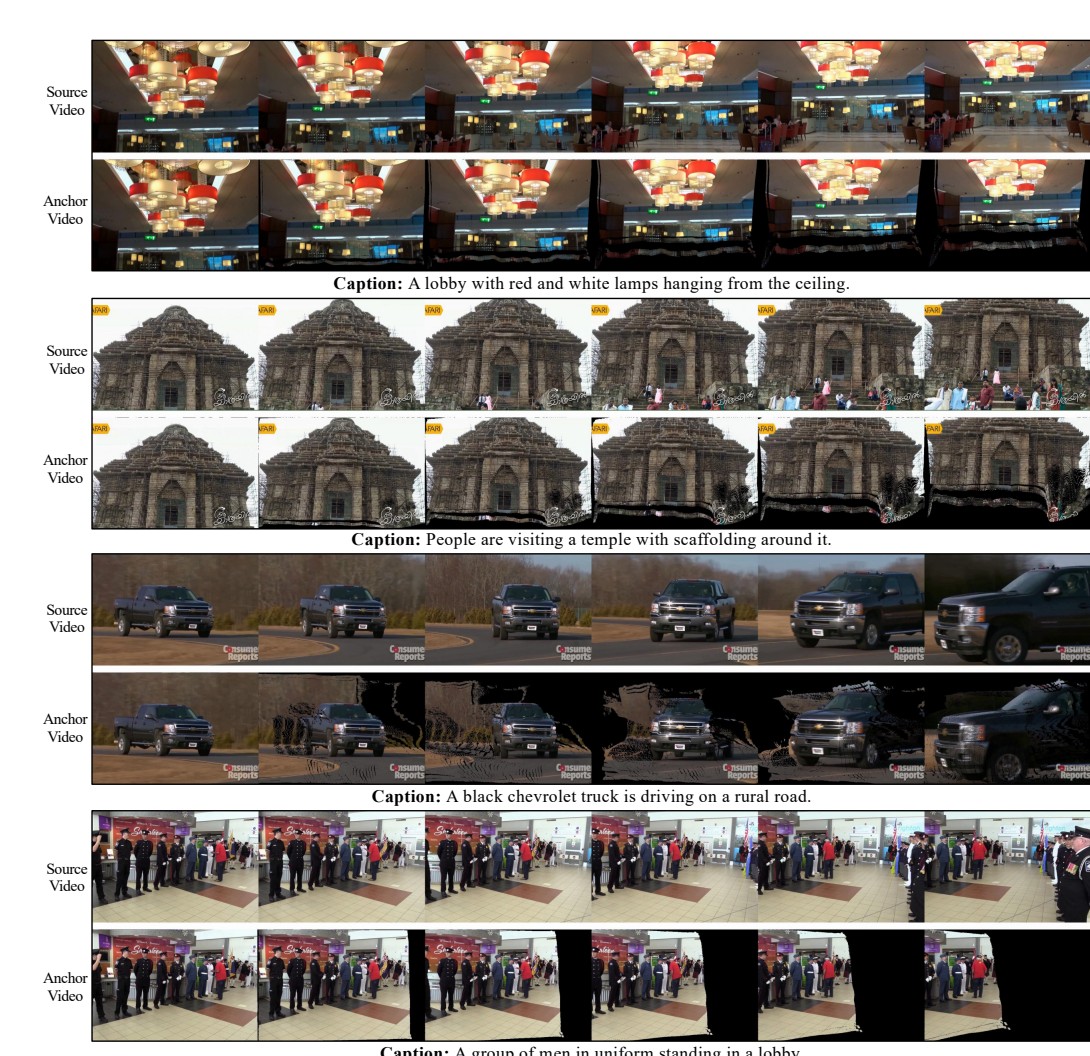

Figure 22: Examples of constructed anchor videos. The source video and corresponding captions are obtained from Panda70M.

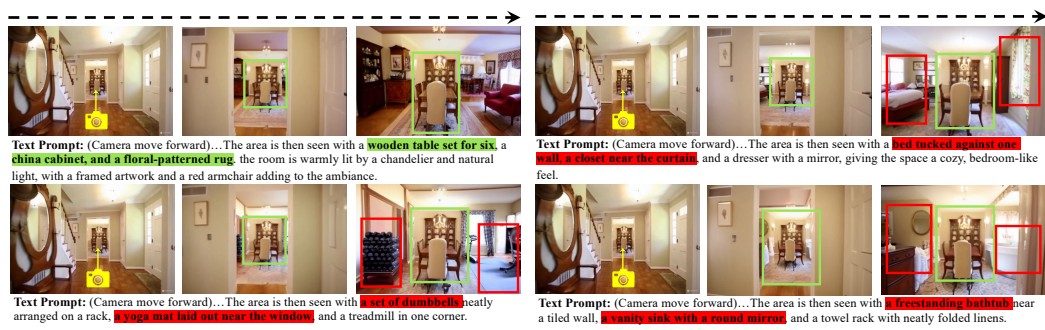

Figure 23: Examples of text-guided scene control.

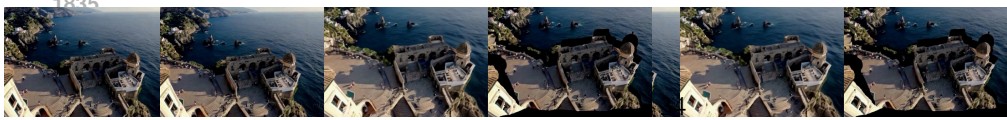

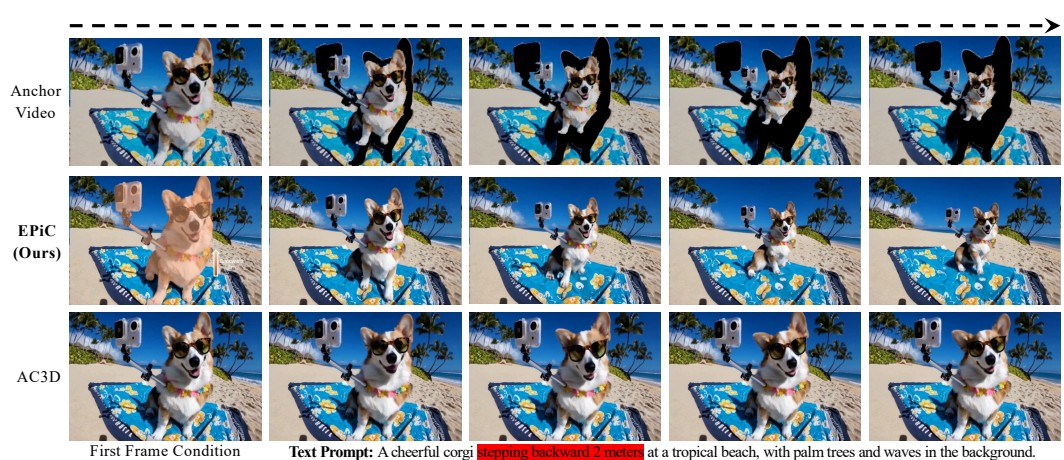

Figure 24: Examples of object 3D trajectory control via anchor video manipulation.

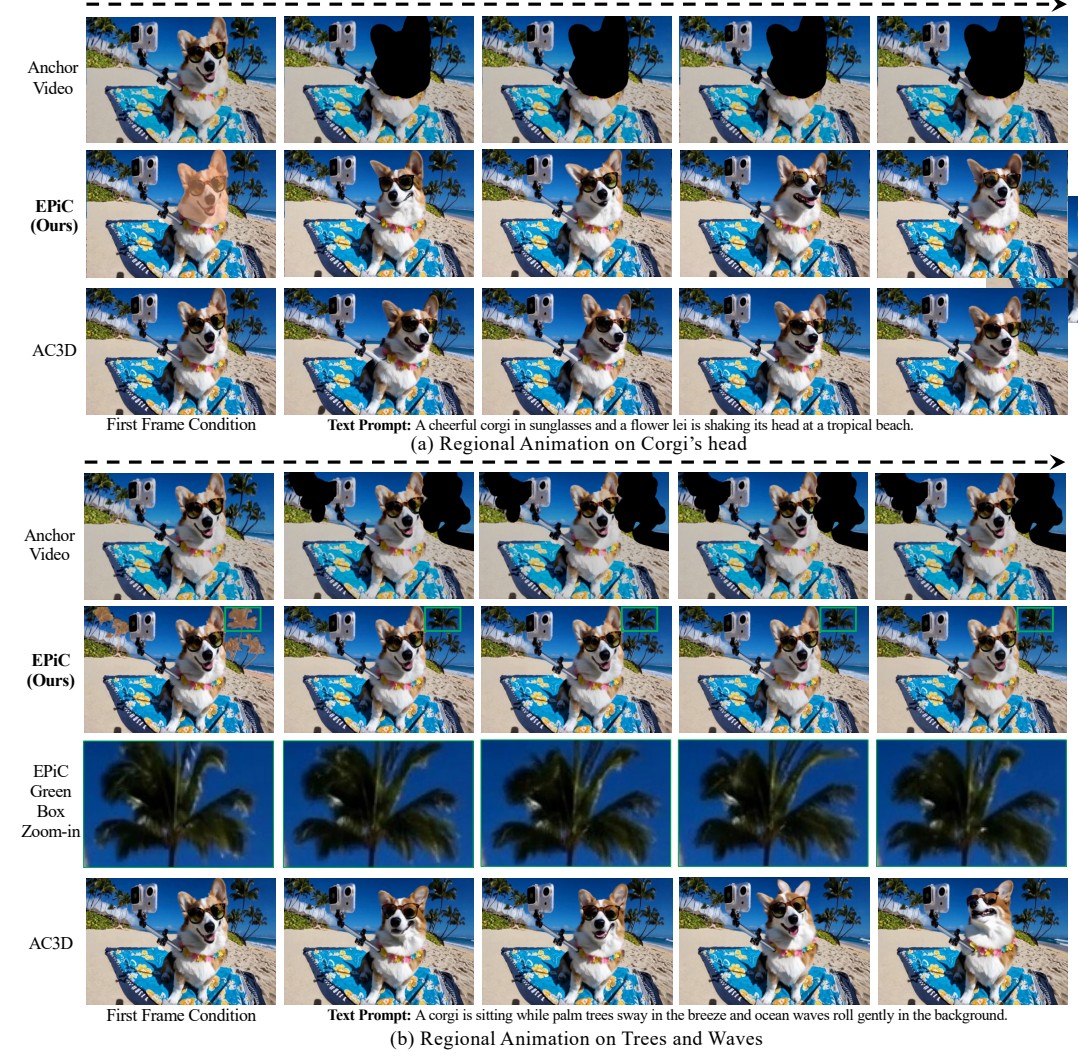

Figure 25: Examples of Regional Animation.

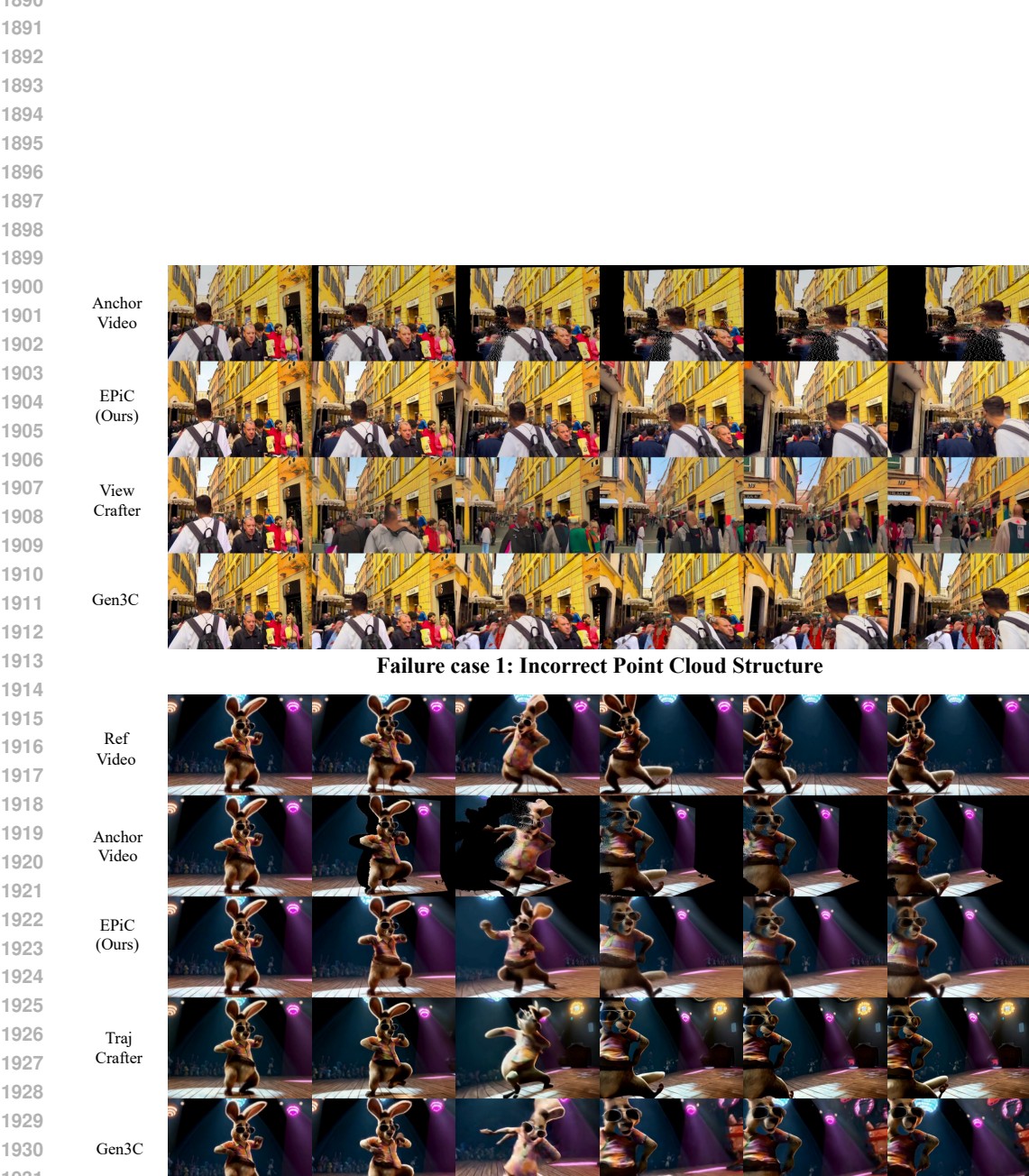

**Failure case 1: Incorrect Point Cloud Structure**

**Failure case 2: Incorrect Occlusion**

Figure 26: EPiC failure cases with baseline comparison.

