# OpenReview forum: "EPiC: Efficient Video Camera Control Learning with Precise Anchor-Video Guidance"
_ICLR.cc/2026/Conference — Submitted to ICLR 2026_

### Official Review · Reviewer_NKTF · 2025-10-17

**Soundness:** 3
**Presentation:** 2
**Contribution:** 3
**Rating:** 2
**Confidence:** 5

**Summary:**

This paper proposes EPiC, a method for camera-controlled video diffusion. Previous methods rely on either Plucker coordinates or depth-based warping techniques using point clouds to condition the video model with camera control. In contrast, this work uses optical flow to create anchor videos to guide the generation process. The paper shows this prevents artifacts of point cloud-based techniques and leads to precise camera control.

**Strengths:**

- Novel conditioning mechanism: Using visibility-based masking with optical flow instead of point-cloud-based conditioning is novel.

**Weaknesses:**

- Short camera trajectories: The demo videos are not very convincing, since the amount of camera motion is very limited and the videos are very short.
- Imprecise camera control: For “Arc Right” in the supplementary website, the bottom result of the third column (city at the coast): the camera continues to move even though it should stop after the first half of rotation. So the camera is not staying fixed at a final pose. The camera seems to continue moving for many cases, especially when the input video has camera motion. So I wonder if there is an issue that the input camera motion of the video is not considered.
- Missing comparisons: FloVD [1] is only briefly discussed, i.e., it does not use anchor videos but uses optical flow. It would have been great to compare with FloVD. Moreover, comparisons with GEN3C [2] as pointcloud-based method are missing. Furthermore, there are no comparisons with ReCamMaster [3] for the task of video-to-video camera retargeting,
- Anchor-ControlNet is nothing novel: Using a lightweight ControlNet to save parameters is pretty standard. For example, AC3D as one of the baselines also does this, if I understand the approach correctly.

[1] Jin et al., FloVD: Optical Flow Meets Video Diffusion Model for Enhanced Camera-Controlled Video Synthesis, CVPR 2025 \
[2] Ren et al., GEN3C: 3D-Informed World-Consistent Video Generation with Precise Camera Control, CVPR 2025 \
[3] Bai et al., ReCamMaster: Camera-Controlled Generative Rendering from A Single Video, ICCV 2025

**Questions:**

While the method is sound, the results are not that convincing. The camera motion is limited and it seems to not be that precise. Moreover, there are some comparisons with recent approaches missing.

I would like authors to address following questions:
- Why is the amount of camera motion that limited? Because of the base video model?
- Why does the camera continue moving for many video-to-video cases even though it says to be fixed after the first movement?
- Why were recent methods like GEN3C or ReCamMaster not used for comparisons?

I think that the paper makes sense but the results and missing comparisons lead to a negative rating. I am happy to consider a rebuttal and open to adjusting my score, though it might be unlikely to accept the paper.

---

> ### Author Response · Authors · 2025-11-24
> **Response (1/4)**
>
> We sincerely thank the reviewer for recognizing the novelty of our visibility-based conditioning mechanism and acknowledging that our proposed method is sound and makes sense. We also really appreciate the reviewer’s expertise in the fields and the constructive feedback on comparisons and evaluation.
>
> ---
> > Clarification on Contribution & Efficiency.
>
> We respectfully re-highlight that one of the primary contributions of EPiC the reviewer may accidentally overlook, is its **exceptional training efficiency** for camera-control learning, which is also recognized by reviewer cqiR, uFzJ and yZGq. This efficiency is crucial because it fundamentally lowers the barrier to precise camera control: instead of requiring industrial-scale compute and massive curated datasets, it enables high-quality camera-controllable video generation training with only modest resources.
>
> As clearly described in our teaser Figure 1 and L47–L53, whereas recent methods such as ViewCrafter, TrajectoryCrafter, etc rely on heavy full-network finetuning and typically consume 1,000+ GPU hours on large-scale datasets, EPiC achieves comparable, and often superior performance on both I2V and V2V benchmarks using only **16 GPU hours and 5,000 videos**. This constitutes a **>60×** reduction in compute and data, made possible by our visibility-based masking strategy and the lightweight Anchor-ControlNet. This highlights that our contribution goes beyond a conditioning or architectural tweak—it represents a meaningful advance in data- and compute-efficient camera-control learning.
>
> ---
> We address the reviewer’s specific concerns as follows. To visually resolve the specific concerns regarding motion scale (W1) and missing comparisons (W3), we also prepared examples in the anonymous website [here](https://epic-iclr-submission.netlify.app/rebuttal.html#r4).
>
> ---
> > W1Q1. Short Trajectories & Limited Motion.
>
> We thank the reviewer for the careful inspection of our demo page. We clarify that the initial I2V demos prioritized scene diversity (using MiraData) over motion magnitude. To address this concern, we have added new examples on the website [R4 Section 1](https://epic-iclr-submission.netlify.app/rebuttal.html#r4-s1), including large-scale rotations and substantial forward/backward translations from both RealEstate10K and MiraData. We believe these examples have a comparable scale of camera motions with prior methods' demo videos.
>
> Regarding video length and its connection to the base model, the reviewer is correct that EPiC inherits the temporal limit (49 frames) of the frozen CogVideoX backbone. However, this is not a limitation specific to our method, but **a general constraint of current video-diffusion backbones**, which are typically trained on short clips. All recent strong baselines cited by the reviewer such as AC3D, FloVD, and Gen3C, are similarly restricted by the temporal range of their underlying base models, and the motion magnitudes shown in our examples are comparable to theirs. We see extending controllable camera motion to longer horizons as an exciting direction for future work, and EPiC can naturally benefit once longer-horizon backbones become available.
>
> ---
> > W2Q2. V2V Camera Control with Moving-Camera Videos
>
> We appreciate the reviewer checking our videos in detail. The reviewer noted that the camera continues moving after the control signal stops. We apologize for any confusion caused by our phrasing “camera stays fixed at the final pose.” We meant that the applied camera pose remains similar to the final pose of the first 1/3 frames, rather than fixed throughout the remaining video. EPiC applies **relative** transformations on top of the source motion. For instance, if the original camera moves forward and the applied control moves left, the resulting motion becomes “forward + left.” Once the control ends, the camera naturally continues with the original forward motion instead of stopping.
>
> This happens because we estimate depth for each frame, lift it to a point cloud, and warp it using the frame’s camera pose. If the source video contains inter-frame motion, our rendered anchor frames will reflect that original motion plus the applied relative control. Thus, in the “city at the coast” example (“Arc Right”, (4,1)-grid video), the camera returns to the original left-rotation trajectory after the control finishes.
>
> Crucially, this design choice **enables us to apply camera control to dynamic, moving-camera videos, a scenario where absolute camera control methods like ReCamMaster often fail**.  As shown on the website [R4 Section 3](https://epic-iclr-submission.netlify.app/rebuttal.html#r4-s3), the suv-in-the-dust and happy-cat examples, ReCamMaster produces static object motion once the camera pose stops changing—contradicting the source video where the objects continue to move. In contrast, EPiC’s relative-pose formulation naturally preserves the original object motion while applying the intended camera control.

---

> ### Author Response · Authors · 2025-11-24
> **Response (2/4)**
>
> >W3Q3. Comparison with FloVD, Gen3C and Recammaster
>
> We thank the reviewer for suggesting these comparisons. We totally agree that these comparisons could ever enhance our experiments and we have performed full evaluations against FloVD, Gen3C, and ReCamMaster below.
>
> ---
> > W3Q3.1 **Quantitative Comparison**
>
> We provide  efficiency/supported task table for EPiC, FloVD, Gen3C and Recammaster, while video quality and camera accuracy/robustness table for for EPiC, FloVD, Gen3C on I2V task.
>
> As shown in the tables below, The results strongly highlight EPiC's advantages in efficiency, precision, and versatility. Specifically:
> - EPiC is the most efficient method: it uses only 5K videos and 16 GPU hours, which is at least **60×–300×** more efficient than prior methods (FloVD: 1500+ GPUh; Gen3C: 5000+ GPUh; ReCamMaster: 600 GPUh).
> - EPiC supports the widest range of tasks, covering both I2V and V2V, and uniquely enables control of generating both static and dynamic regions, whereas others can control only one type.
> - EPiC’s video quality is comparable or better than state-of-the-art baselines on both RealEstate10K and MiraData across all metrics.
> - EPiC achieves the most accurate and robust camera control, thanks to its strong anchor video following ability.
>
> **Efficiency-Supported Task Table**
> | Model        | Training Data | GPU Hours | Parameters | Supported Tasks | I2V Controllable Region Dynamics |
> |--------------|----------------|-----------|------------|------------------|-------------------------------|
> | FloVD        | 600K           | 1500+     | 1.4B       | I2V             | Only dynamic                  |
> | Gen3C        | >100K          | >5000     | 7.23B      | I2V / V2V       | Only static                   |
> | RecamMaster  | 136K           | 600       | 1.49B      | V2V             | —                             |
> | Ours         | **5K**             | **16**        | **26M**        | **I2V / V2V**       | **Static/dynamic**                |
>
> **Video Quality Table**
> | Dataset | Method        | Total | Subject Consist | Bg Consist | Motion Smooth | Temporal Flicker | Aesthetic Quality | Imaging Quality |
> |---------|---------------|-------|------------------|-------------|----------------|-------------------|--------------------|------------------|
> | RE10K   | FloVD         | **82.68** | **91.77**       | 93.25      | 98.30         | 96.23            | 50.97             | **65.56**       |
> | RE10K   | Gen3C         | 82.27 | 91.10           | 92.75      | 97.99         | **96.67**        | 50.61             | 64.54           |
> | RE10K   | EPiC (Ours)   | 82.63 | 91.62           | **93.43**  | **98.48**     | 96.47            | **51.19**         | 64.57           |
> | MIRA    | FloVD         | 82.55 | 91.64           | 92.91      | 98.43         | **94.67**        | 57.46             | 60.21           |
> | MIRA    | Gen3C         | 80.50 | 88.56           | 90.75      | 96.76         | 91.74            | 55.21             | 59.98           |
> | MIRA    | EPiC (Ours)   | **82.89** | **91.82**   | **92.94**  | **98.75**     | 94.86            | **57.94**         | **61.03**       |
>
> **Camera Control Accuracy / Robustness Table**
> | Dataset | Method       | Rotation Error (↓) | Transition Error (↓) | CamMC (↓)     |
> |---------|--------------|---------------------|------------------------|----------------|
> | RE10K   | FloVD        | 0.76 ± 0.31         | 1.14 ± 0.52            | 1.47 ± 0.56    |
> | RE10K   | Gen3C        | 0.45 ± 0.13         | 0.99 ± 0.22            | 1.35 ± 0.30    |
> | RE10K   | EPiC (Ours)  | **0.40 ± 0.11**     | **0.86 ± 0.18**        | **1.17 ± 0.23**|
> | MIRA    | FloVD        | 0.95 ± 0.44         | 2.15 ± 0.98            | 3.48 ± 1.03    |
> | MIRA    | Gen3C        | 0.81 ± 0.24         | 2.05 ± 0.77            | 2.75 ± 0.72    |
> | MIRA    | EPiC (Ours)  | **0.66 ± 0.22**     | **1.78 ± 0.67**        | **2.10 ± 0.60**|

---

> ### Author Response · Authors · 2025-11-24
> **Response (3/4)**
>
> > W3Q3.2 **Qualitative Comparisons**
>
> We provide qualitative comparisons to baseline methods on the website [R4 Section 2 ](https://epic-iclr-submission.netlify.app/rebuttal.html#r4-s2) and [R4 Section 3](https://epic-iclr-submission.netlify.app/rebuttal.html#r4-s3).
>
> **Controllable Dynamic Objects in I2V**. As shown in the examples on the website [R4 Section 2.1 ](https://epic-iclr-submission.netlify.app/rebuttal.html#r4-s2-1), EPiC flexibly supports both **dynamic and static** scenes in I2V. By contrast, FloVD mainly handles dynamic objects, and Gen3C supports only static scenes. EPiC can naturally do both by simply adjusting the mask in the anchor video to specify which regions should move and which should stay fixed.
>
> **Qualitative Comparisons With FloVD**. We provide qualitative comparisons with FloVD on the website [R4 Section 2.2](https://epic-iclr-submission.netlify.app/rebuttal.html#r4-s2-2). FloVD and EPiC show comparable visual quality on both MiraData and RealEstate10K, but EPiC achieves **more accurate camera motion and is much more robust to random seeds**.
>
> We attribute this difference to training alignment. FloVD relies on noisy estimated optical flow for training, causing input–output misalignment and unstable flow following, which leads to higher variance and larger camera-control errors. EPiC instead uses visibility-based masking to enforce pixel-level input-output alignment and clean supervision, resulting in more stable, accurate, and seed-robust camera control.
>
> **Qualitative Comparisons With Gen3C**. We provide qualitative comparisons with Gen3C on the website [R4 Section 2.3](https://epic-iclr-submission.netlify.app/rebuttal.html#r4-s2-3). Both ours and Gen3C can follow the anchor video reasonably well, while **Gen3C’s visual quality is noticeably lower than ours on MiraData**. We suspect this is because Gen3C is trained primarily on scene-centric datasets, and its full finetuning process tends to turn the base model into a scene-level NVS model, which struggles to handle human-involved examples effectively.
>
> Since Gen3C also supports V2V, we include a V2V comparison in the third video. Gen3C tends to follow the anchor video too strictly, reproducing the same incorrect occlusion artifacts. In contrast, **EPiC generates more reasonable content thanks to preserving the backbone’s first-frame semantic prior**.
>
> **Qualitative Comparisons with Recammaster**. We provide qualitative comparisons with Recammaster on the website [R4 Section 3](https://epic-iclr-submission.netlify.app/rebuttal.html#r4-s3).
>
> Compared to EPiC, we observe the following issues with ReCamMaster:
> - Failure on moving-camera videos: ReCamMaster often makes both the camera and objects static in the last frames when the source video also contains camera motion (e.g., suv-in-the-dust, happy-cat), contradicting the source video where the objects continue to move.
> - Structural inconsistencies: Without explicit 3D guidance like EPiC’s anchor video, it can distort scene geometry (e.g., basketball-explosion, vlogger-corgi, photoreal-train).
> - Hallucinations and artifacts: ReCamMaster may introduce non-existent objects or produce flickering / oil-paint-like artifacts (e.g., basketball-explosion, art-museum).
>
> In contrast, with explicit 3D guidance and the backbone’s semantic prior, EPiC handles these cases reliably.
>
> We also find ReCamMaster cannot reproduce the same camera trajectory across seeds as shown in the website [R4 Section 3.2](https://epic-iclr-submission.netlify.app/rebuttal.html#r4-s3-2), while EPiC maintains consistent trajectories.

---

> ### Author Response · Authors · 2025-11-24
> **Response (4/4)**
>
> > W4. Novelty of Anchor-ControlNet.
>
> We appreciate the reviewer’s thoughtful questions. We clarify that being “lightweight” is only a byproduct—not our motivation to use ControlNet. We use ControlNet architecture because (1) the anchor video is a low-level signal, which can be pretty suitable for ControlNet to model if anchor–source alignment is preserved, and (2) freezing the backbone preserves the first-frame semantic prior.
>
> Thus, unlike existing methods with achor video guidance (TrajCrafter, Gen3C, and ViewCrafter), which (almost) fully finetune the backbone on large-scale datasets, we treat the anchor video purely as a control signal and learn it through a small ControlNet. More importantly, we introduce a **masking-based regional injection**: the ControlNet output is added only to visible regions, simplifying learning to a copy-paste problem and improving quality (original paper L94-L99; Fig. 5(c) ). **This also enables EPiC to generalize to different masked anchor videos at test time (Fig. 2), supporting both static and dynamic scenes with user-specified dynamic regions** (examples in website [R3 section1](https://epic-iclr-submission.netlify.app/rebuttal.html#r3-s1) and [R4 Section2.1](https://epic-iclr-submission.netlify.app/rebuttal.html#r4-s2-1)).
>
> While AC3D also uses ControlNet, it applies it only to camera parameters and does not involve specific design like our masking-based anchor conditioning, and it targets T2V rather than I2V/V2V. Thus, its setting and design goals are different from ours.
>
> Overall, our masking-based Anchor-ControlNet is a novel design that fundamentally simplifies learning, preserves semantic priors, and provides region-level controllability that prior methods do not support.

---

> ### Comment · Reviewer_NKTF · 2025-11-27
>
> Thanks for the thorough rebuttal. The additional results help a lot and the clarification about relative camera control w.r.t. the input is important. Please incorporate all this in the paper. My only remaining concern is that the Anchor-ControlNet being a contribution is an overclaim, I do not think this is a contribution, as it has been done before and just adapted to a different input without any challenges required to adapting. Considering the rebuttal, I would be fine seeing the paper accepted, assuming everything will be incorporated.

---

> ### Author Response · Authors · 2025-12-01
> **Thank You for the Positive Feedback and Support for Acceptance**
>
> We sincerely thank the reviewer for their encouraging feedback. We are glad that our thorough rebuttal and additional results were helpful, and we greatly appreciate the indication that **the paper is now considered acceptable for publication**.
>
> ---
> > **Clarification on Anchor-ControlNet Design**
>
> Regarding the Anchor-ControlNet contribution, we want to clarify that **simply using an anchor video as a ControlNet input signal is not trivial**, specifically due to the partial visibility challenge inherent in the data (since the anchor video is not fully aligned with the source video, unlike classical low-level control signals). As a result, as shown in Fig. 5(c) and on our website [R4 Section 4](https://epic-iclr-submission.netlify.app/rebuttal.html#r4-s4), **directly applying a vanilla ControlNet to the anchor video (without masking)  (2nd column) causes the model to follow erroneous invisible regions**, resulting in black or distorted content (in the first row the model follows the left-back region, while in the second row it follows the left edge-stretching artifacts). This shows that a simple ControlNet cannot handle anchor-video conditioning.
>
> Additionally, adding the proposed visibility-aware output masking (VAOM) only during inference (3rd column) also proves insufficient: it still causes flickering in several regions, and invisible areas cannot properly extend the scene (e.g., the first example where a black region becomes a brown patch).
>
> In contrast, using the proposed VAOM during both training and inference fully unlocks the base model’s ability to complete invisible regions naturally and coherently, yielding clean, stable, and artifact-free results. This also enables EPiC to generalize to arbitrary masked anchor videos at test time (Fig. 2), supporting both static and dynamic scenes with user-specified dynamic regions (examples in website [R3 section1](https://epic-iclr-submission.netlify.app/rebuttal.html#r3-s1) and [R4 Section2.1](https://epic-iclr-submission.netlify.app/rebuttal.html#r4-s2-1)).
>
> > **Summary of Anchor-ControlNet Novelty and Contribution**
>
> Overall, we design the first efficient learned-ControlNet architecture tailored for anchor-video–based camera control, where prior methods (TrajCrafter, Gen3C, ViewCrafter) fully finetune the backbone on large datasets. Moreover, **our region-based, visibility-aware output masking design enables natural visible region completion, as well as controllable regional motion**, a capability that has not been explored in prior camera control or video generation work. In particular, to the best of our knowledge, no existing ControlNet variants address region-specific conditioning under partial visibility, making our design the first to solve this problem.
>
> We believe this constitutes a meaningful and novel contribution to the camera control field.
>
> ---
> Thanks again for your valuable contributions throughout the review and discussion period. We sincerely appreciate your expertise and the thoughtful questions you raised, which have greatly helped strengthen our paper. **We’ve incorporated all the discussed contents during rebuttal into the revised paper (Updated texts are shown in blue), as well as updated the main website with these additional results and comparisons.**

---

### Official Review · Reviewer_uFzJ · 2025-10-18

**Soundness:** 3
**Presentation:** 3
**Contribution:** 3
**Rating:** 6
**Confidence:** 3

**Summary:**

This paper presents EPiC (Efficient and Precise Camera Control), a framework for efficient learning of camera motion control in video diffusion models (VDMs). Instead of relying on point-cloud rendering and camera trajectory estimation—which often introduce pixel-level misalignment—the authors propose a visibility-based masking strategy to construct well-aligned anchor videos directly from source videos. They further introduce a lightweight Anchor-ControlNet, accounting for less than 1% of the backbone parameters, which conditions video generation on anchor videos through visibility-aware output masking. EPiC enables efficient training (only 5K videos and 500 iterations) and achieves state-of-the-art camera control accuracy on RealEstate10K and MiraData, while also demonstrating strong zero-shot generalization from image-to-video (I2V) to video-to-video (V2V) tasks. Extensive quantitative, qualitative, and ablation studies validate the proposed approach’s efficiency, robustness, and alignment advantages over existing baselines such as CameraCtrl, AC3D, and ViewCrafter.

**Strengths:**

1. Clear and interpretable design.
2. The idea of ​​converting the problem into anchor-video construction is very interesting.

**Weaknesses:**

1. Converting camera-guided video generation into the task of supplementing an anchor video is an interesting idea. However, when applied to I2V, although the authors preserve foreground dynamics by masking foreground regions during guidance, the background is effectively forced to remain static. This imposes a limitation when treating video generation as a world model. Moreover, the approach depends to some extent on the performance of foreground extraction models (e.g., *GroundingDINO* used in the paper), which requires users to provide additional inputs.

2. In the Introduction, the authors state that the Anchor-ControlNet is “injected into the first 25% of backbone layers”, while the original ControlNet is injected into 50% of layers. What specific design or empirical analysis motivated selecting 25%? Some works (e.g., *InstantStyle*) experimentally show that particular layers predominantly control generation style, and therefore only control those layers. Did the authors observe similar layer-wise effects? Please provide more design details and/or empirical evidence supporting the 25% choice.

3. Minor :

   a. Line 76: change “We propose” → “we propose” (lowercase “we”).
   b. Make figure references consistent throughout the manuscript; do not mix full form and abbreviation. For example, Line 367 uses “Figure 1” while Line 375 uses “Fig. 4”. Please standardize notation across the paper.

**Questions:**

See Weakness.

---

> ### Author Response · Authors · 2025-11-24
>
> We thank the reviewer for finding our core idea of anchor-video construction "very interesting" and our design "clear and interpretable." We appreciate the recognition of our extreme efficiency (requiring <1% parameters and 500 iterations), SOTA accuracy on RealEstate10K and MiraData, and strong zero-shot generalization across I2V and V2V tasks. We address the specific concerns below. Several qualitative examples supporting our reply are provided on the website [here](https://epic-iclr-submission.netlify.app/rebuttal.html#r3).
>
> ---
> > W1.1  foreground/background masking
>
> We thank the reviewer for the insightful observation. Our masking strategy is not limited to foreground regions: users may also designate some background areas as dynamic by masking background regions in the anchor video. We provide three examples to illustrate this on the website [here](https://epic-iclr-submission.netlify.app/rebuttal.html#r3-s1) (Dynamic Background Control). By masking background regions, EPiC can also generate dynamic backgrounds (Example 1: flickering neon lights; Example 2: rippling river water; Example 3: swaying trees). Thus, the method is not restricted to foreground motion and supports controllable dynamics in both foreground and background regions.
>
> ---
> W1.2 Masking Method Flexibility
>
> We actually mentioned in L306 that “users may also customize tailored segmentation masks,” indicating that GroundedSAM is not definitely required. In fact, these masks only specify where the model is allowed to freely generate without anchor-video guidance, so they do not need to be semantic masks at all. Users can simply mark dynamic regions with manual brushes, and we provide such examples in the “Manually Brushed Mask” examples on the website [here](https://epic-iclr-submission.netlify.app/rebuttal.html#r3-s1). As shown, we can animate the corgi’s head or the tree leaves by manually masking those regions. Thus, EPiC does not depend on any specific segmentation model and supports flexible, user-defined control over dynamic regions.
>
> ---
> W2 Controlnet Injecting Layers
>
> We thank the reviewer for raising such detailed question. As shown in our additional ablations (Appendix Table 5 in the original pdf, Table 7 in the revised pdf), extending ControlNet injection beyond the first 25% does not improve camera-control accuracy. We hypothesize that camera motion is a coarse, sketch-level signal; shallow layers are sufficient to impose trajectory guidance, while deeper layers mainly refine appearance and semantics after the camera motion has already been fixed. Therefore, the 25% choice provides the best trade-off between effectiveness and stability.
>
>
> ---
> W3 Style/typo consistency
>
> Thank you for pointing this out. We have fixed these in the updated paper pdf.

---

### Official Review · Reviewer_yZGq · 2025-10-31

**Soundness:** 3
**Presentation:** 3
**Contribution:** 3
**Rating:** 4
**Confidence:** 3

**Summary:**

This paper proposes EPiC, an efficient framework for camera control learning in video diffusion models. It constructs well-aligned anchor videos through first-frame visibility masking. The authors further design a lightweight Anchor-ControlNet architecture with visibility-aware output masking. EPiC achieves state-of-the-art camera control accuracy while requiring only a fraction of the training cost compared to prior methods.

**Strengths:**

1.	The visibility-based masking for anchor video generation is conceptually simple yet effective.
2.	The method achieves SOTA quality and camera control scores on both I2V and V2V test sets, and the ablation studies verify the effectiveness of the proposed methods.

**Weaknesses:**

1. The principal contribution lies in constructing more precisely aligned anchor videos to improve training efficiency. Is the proposed module plug-and-play, and could it likewise enhance other diffusion-based video models?
2. I recommend including a failure-case analysis to more thoroughly illustrate the method’s limitations.
3. During inference, how robust is EPiC to substantial errors in point-cloud-rendered anchors? Since it is trained with high-quality mask-based anchors, this discrepancy could introduce a train–test mismatch.

**Questions:**

Do the authors plan to release the mask-based anchor-video training dataset? Its availability would greatly benefit the research community.

---

> ### Author Response · Authors · 2025-11-24
>
> We thank the reviewer for finding our visibility-based masking 'conceptually simple yet effective' and acknowledging our SOTA performance on both I2V and V2V tasks. We also appreciate the recognition of our efficiency (requiring only a fraction of training cost) and the validity of our ablation studies. We address the specific concerns below. Several qualitative examples supporting our reply are provided on the website [here](https://epic-iclr-submission.netlify.app/rebuttal.html#r2).
>
> ---
> >W1.  Is EPiC plug-and-play for other video diffusion backbones?
>
> Yes. The Anchor-ControlNet and our masking-based anchor-video construction are both backbone-agnostic and do not rely on any CogVideoX-specific components.
>
> To demonstrate this, we applied EPiC with the same settings on the Wan2.1-I2V-14B-480P backbone by training a lightweight masking-based Anchor-ControlNet on top. We report both quantitative and qualitative results on our collected RealEstate10K test set below.
>
> | Method           | Total | Subject Consist | Bg Consist | Motion Smooth | Temporal Flicker | Aesthetic Quality | Imaging Quality |
> |------------------|-------|------------------|-------------|----------------|-------------------|--------------------|------------------|
> | EPiC+CogVideoX   | 82.63 | 91.62           | 93.43      | 98.48         | 96.47            | 51.19             | 64.57           |
> | EPiC+Wan2.1      | 84.24 | 92.97           | 93.54      | 98.53         | 97.42            | 55.67             | 67.34           |
>
> | Method         | Rotation Error (↓) | Transition Error (↓) | CamMC (↓)     |
> |----------------|---------------------|------------------------|----------------|
> | EPiC+CogVideoX | 0.40 ± 0.11         | 0.86 ± 0.18            | 1.17 ± 0.23    |
> | EPiC+Wan2.1    | 0.41 ± 0.10         | 0.84 ± 0.20            | 1.15 ± 0.21    |
>
>
> We also provide qualitative examples on the website [here](https://epic-iclr-submission.netlify.app/rebuttal.html#r2-s1), showing consistent anchor-video following ability.
>
> This confirms that our design is indeed plug-and-play and can enhance a broad family of video diffusion models without architectural modifications.
>
> ---
> > W2. Failure case analysis
>
> We agree that failure analysis is important. Since our model learns to follow the anchor video in visible regions, it can be affected when the estimated point-cloud structure or occlusion masks are inaccurate. We provide two examples on the website [here](https://epic-iclr-submission.netlify.app/rebuttal.html#r2-s2), illustrating the main failure modes:
>
> - Incorrect point-cloud structure. In the first example, a misestimated point cloud causes the person in the anchor video to appear tilted, and our result partially inherits this (e.g., a slightly stretched neck). However, unlike ViewCrafter—which loses track of the motion and produces heavily distorted bodies—EPiC remains noticeably more stable.
>
> - Incorrect occlusion. In the second example, background color leaks through the kangaroo’s face in the anchor video. EPiC converts this into a mild blue lighting effect, whereas TrajectoryCrafter rigidly copies the artifact and produces visible holes in the face.
>
> These analyses clarify how EPiC behaves under imperfect 3D estimation and demonstrate that, even in failure cases, it remains more robust than baseline methods.
>
> ---
> > W3. Robustness to point-cloud errors at inference
>
> As mentioned above, while EPiC can be affected by point-cloud inaccuracies as mentioned above, our results remain noticeably more natural than baselines. This is because EPiC keeps the backbone fully frozen, preserving a strong first-frame semantic prior. Thus, guidance comes jointly from the anchor video and the first-frame semantics, enabling EPiC to follow the anchor while avoiding implausible artifacts.
>
> We’ve already shown  the above two examples to demonstrate this.  We also have another in the main paper Fig. 4(b), where EPiC removes the leaked tree artifact from the corgi’s face thanks to the backbone prior, but TrajectoryCrafter copies it.
>
> These results show that EPiC is significantly more robust to point-cloud errors than competing methods. Importantly, since EPiC can work with any 3D estimator (we already use different depth models for I2V and V2V), its performance can also naturally improve as the underlying 3D estimation improves with stronger modern 3D estimation models.
>
> ---
> > Q1. Will the mask-based anchor-video dataset be released?
>
> Yes. We have included the source code for computing visibility masks using estimated optical flow in the supplementary materials, which can be directly applied to open-source video datasets such as OpenVid-1M and Panda-70M. We will also release the processed latents and videos used in our experiments in the final version.

---

### Official Review · Reviewer_cqiR · 2025-10-31

**Soundness:** 3
**Presentation:** 3
**Contribution:** 3
**Rating:** 6
**Confidence:** 4

**Summary:**

This paper introduces EPiC, a novel framework for efficient camera control in video diffusion models. It replaces error-prone 3D point clouds with a new method for creating "anchor videos" by directly masking source videos based on optical flow. This produces perfectly aligned training data from any video, enabling remarkable efficiency. Combined with a lightweight Anchor-ControlNet, EPiC achieves state-of-the-art results with a fraction of the data and compute of previous methods, and generalizes well to video-to-video tasks.

**Strengths:**

1. The core strength is the visibility-masking method for creating aligned anchor videos. This elegantly bypasses the need for 3D reconstruction, enabling more efficient training in data and compute than prior work.
2. EPiC achieves SOTA results on standard I2V camera control benchmarks (RealEstate10K, MiraData), demonstrating that the efficiency gains do not compromise final quality.
3. The model shows excellent zero-shot generalization to V2V tasks and demonstrates strong capability in handling dynamic scenes, a key advantage over methods trained primarily on static data.

**Weaknesses:**

1. The framework operates on a relative scale determined by an external depth estimator at inference time. This may prevents users from specifying precise, real-world camera movements (e.g., "move 2 meters").
2. While qualitative results are promising, the paper lacks quantitative validation on a benchmark with dynamic objects and ground-truth camera motion (e.g., RealCam-Vid).  This would allow for a direct and fair comparison of dynamic handling capabilities with methods like RealCam-I2V.

**Questions:**

1. The interactive interface for drawing trajectories is a strong feature. Is the motion scale purely relative to the scene's estimated depth, or is there a mechanism to map it to an absolute, metric scale for consistent control?
2. How robust is the training data generation to challenging scenarios where optical flow often fails, such as very fast camera motions (e.g., drone footage) or large-scale rotations?

---

> ### Author Response · Authors · 2025-11-24
>
> We thank the reviewer for recognizing the novelty and efficiency of our visibility-masking framework, our SOTA performance on I2V camera control benchmarks, and our model's strong capability in handling dynamic scenes and zero-shot V2V tasks. We also appreciate the positive feedback on our interactive trajectory interface. We address the reviewer's concerns as follows.
>
> ---
> > W1Q1. Relative vs. Metric Camera Scale
>
> Thank you for your concern, and we’d like to clarify that the camera scale in our work is not purely relative. In the I2V setting, our inference-time depth is obtained from a metric depth estimator (Depth Anything v2; original pdf L296–299, revised pdf L307–308), which allows users to specify absolute real-world camera displacements, such as “move camera forward with 2 meters”.
>
> Moreover, we already include an example in the original version of our paper, demonstrating explicit metric-scale object-motion control in our 3D object-trajectory control application (“corgi steps backward 2 meters”), shown in Fig. 20 (original pdf, Fig. 24 in revised pdf) in the Appendix. In this example, we compute metric depth, unproject pixels into 3D, translate the corgi’s 3D points backward by 2 meters, and then render both the anchor video and the synthesized output. This confirms that our framework supports metric-scale motion control.
>
> ---
>  > W2.  RealCam-Vid Evaluation
>
> We sincerely appreciate the reviewer’s acknowledgment of our promising qualitative results. We would also like to respectfully clarify that we have already evaluated our method on RealCam-Vid, as presented in Lines 320–323 (original pdf, L335-338 in revised pdf). As running the full RealCam-I2V test set (5,000 clips) requires ≈2,500 GPU-hours,  we follow a practical protocol, where we sample 1,000 clips from RealCam-I2V (500 RealEstate10K (mostly static scenes) + 500 MiraData (scenes with dynamic objects, which is the only source with dynamic objects in RealCam-Vid). This selection gives us diverse static + dynamic scenes, which we believe is sufficient to evaluate the reviewer’s concerns about dynamic handling, and the results (Tab. 1) demonstrate our superior performance against others.
>
> ---
> > Q2.  Robustness of Training Data Generation
>
> We agree that optical flow can be inaccurate in challenging scenarios. However, our method does not rely on precise flow estimation. As mentioned in L78-80, our key idea is to ensure pixel-level alignment between the anchor and source video, and achieving this doesn’t require the visibility mask derived from optical flow to be perfect, as we directly mask the source. In practice, we observe that flow inaccuracies do not propagate into the training signal (the loss remains stable, as shown in Fig. 5(a)). With our masking-based ControlNet design, the model only receives guidance from reliable visible regions (where anchor-video is aligned with source video), turning the learning process into a simple copy-paste operation. Consequently, training remains robust despite imperfect flow.

---

### Author Response · Authors · 2025-12-03
**AC Note: Summary of Rebuttal and Discussion (1/2)**

Dear AC,

We sincerely appreciate your time and effort in meta-reviewing our paper.

# Reviewers' Positive Feedback

We are grateful to the reviewers for their careful evaluation and insightful suggestions. We summarize the advantages of our method recognized by the reviewers:

- **Novel visibility-based masking for constructing well-aligned anchor videos**, avoiding point-cloud errors and 3D reconstruction issues (cqiR, yZGq, uFzJ, NKTF).
- **Lightweight Anchor-ControlNet design with visibility-aware output masking**, adding <1% parameters while enabling effective conditioning (yZGq, uFzJ).
- **Major efficiency gains**, achieving strong results with only **5K videos, 26M parameters and 16 GPU hours** — over **50×** more data-, parameter- and compute-efficient than prior work (cqiR, yZGq, uFzJ).
- **State-of-the-art I2V camera control accuracy and video quality** on both static and dynamic scenarios such as RealEstate10K and MiraData (cqiR, yZGq, uFzJ).
- **Strong zero-shot generalization to V2V tasks**, including robust handling of dynamic scenes (cqiR, yZGq, uFzJ).
- **Improved robustness and precision**, with **effective prevention** of point-cloud artifacts (cqiR, uFzJ, NKTF).



# Major Responses and Revisions

During the rebuttal period, we thoroughly addressed all reviewer questions. We are encouraged that three **reviewers (cqiR, yZGq, uFzJ) initially rated our paper as **Soundness: 3 (good), Presentation: 3 (good), Contribution: 3 (good)** and provided positive assessments of our paper**. Additionally, the remaining reviewer NKTF (initial score 2, confidence 5) acknowledged that our additional results, comparisons, and clarifications in rebuttal satisfactorily resolved their questions, stating: **“Considering the rebuttal, I would be fine seeing the paper accepted, assuming everything will be incorporated.”**, indicating that their issues have been effectively addressed and that they are now supportive of acceptance. We summarize the reviewers’ questions and how we addressed each of them below.

> **Reviewer NKTF's questions**:

- Not large camera motions in the demos.

    **Response**: We added extensive large-camera motion examples on our [rebuttal website](https://epic-iclr-submission.netlify.app/rebuttal.html#r4-s1), demonstrating EPiC’s stability and controllability under strong motion.

- Generated video’s camera is still moving in several V2V examples.

    **Response**: We clarified that EPiC uses **relative** (not absolute) camera trajectories, so the output motion is the **composition** of the source video motion and the applied control. We updated this explanation in the paper (L322–326) and on the website ([link](https://epic-iclr-submission.netlify.app/#v2v-gallery)), and added examples illustrating why relative composition outperforms absolute methods like ReCamMaster ([link](https://epic-iclr-submission.netlify.app/rebuttal.html#r4-s3)).

- Missing comparisons to FloVD, Gen3C, and ReCamMaster.

    **Response**: We added **comprehensive I2V and V2V comparisons with all three methods**, including qualitative results (Fig. 4, 12–16) and quantitative results (Table 1), showing that **EPiC outperforms them in both controllability and stability**.

- Novelty of Anchor-ControlNet.

    **Response**: We explained the challenges of applying ControlNet to anchor-video conditioning and highlighted our novel **visibility-aware output masking**, which prevents invisible-region artifacts and enables selective region control, supported by new ablations (Appendix D.4, Fig. 9).

Importantly, after these clarifications and additional experiments, the reviewer concluded:
**"Considering the rebuttal, I would be fine seeing the paper accepted, assuming everything will be incorporated."**
indicating that their concerns were effectively addressed and they are now supportive of acceptance.

---

> ### Author Response · Authors · 2025-12-03
> **AC Note: Summary of Rebuttal and Discussion (2/2)**
>
> > **Reviewer yZGq's questions**:
>
> - Generalization across different backbones.
>
>     **Response**: We provide **experiments showing EPiC generalizes to the Wan2.1 backbone** (Appendix D.5), with both quantitative (Table 7) and qualitative (Figure 10) results demonstrating consistent performance across architectures.
>
> - Failure-case and robustness analysis regarding point-cloud errors.
>
>     **Response**: We added **detailed failure-case analyses** illustrating how EPiC behaves under point-cloud or anchor-video rendering errors (Appendix I and Figure 26), and showed that EPiC remains **more robust and faithful than baseline methods** even when such errors occur.
>
>
>
> > **Reviewer cqiR's questions**
>
> - The camera scale is not absolute.
>
>     **Response**: Our method **already uses the metric depth estimator** Depth Anything v2 (L307-308), which provides scale-aware depth, ensuring that camera scale is properly handled.
>
> - Evaluation on RealCam-Vid with dynamic scenes.
>
>   **Response**: Our main paper **already includes experiments on the RealCam-Vid subset**, with quantitative results (Table 1) and qualitative examples (Fig. 4, 12–14, 17) on both static and dynamic scenes (RealEstate10K and  MiraData).
>
>
>
>
> > **Reviewer uFzJ's questions**:
> - Reliance on segmentation models and potential restriction to foreground-only motion.
>
>     **Response**: We provided **qualitative examples on our rebuttal website** ([link](https://epic-iclr-submission.netlify.app/rebuttal.html#r3-s1)) using manually drawn masks and dynamic-background masks, demonstrating that EPiC supports flexible, user-defined dynamic regions and is **not limited** to foreground segmentation.
>
> - Need for more ablations on the lightweight Anchor-ControlNet.
>
>     **Response**: We clarified that the lightweight-design ablations are **already included in Appendix D.2**, showing that our **<1%-parameter** Anchor-ControlNet remains effective and stable.
>
> # Summary
>
> In summary, the **majority of reviewers recommend acceptance** and recognize our paper’s core contributions in (1) visibility-based anchor-video construction, (2) lightweight and effective Anchor-ControlNet design, (3) strong efficiency with orders-of-magnitude lower data and compute, and (4) state-of-the-art camera-control accuracy and generalization. Their comments **mainly asked for additional experiments or clarifications, all of which we have thoroughly addressed** in the rebuttal. Additionally, we have **incorporated all the new experiments, examples, and comparisons in both the revised paper as well as our [anonymous website](https://epic-iclr-submission.netlify.app)**.
>
> We hope these clarifications and additional results clearly demonstrate the strength and completeness of our contributions.
> We sincerely thank the AC for the thoughtful handling of our submission.
>
> Best,
>
> Authors

---

### Meta-Review · Area_Chair_DaSj · 2026-01-07

**Summary:**

Initial scores were mixed, from 2 to 6. Most concerns were about whether the method is sound, mainly the gap between training and inference, and about the claimed novelty. Reviewer yZGq pointed out a train test mismatch: training uses clean 2D optical flow masks, but inference relies on noisy 3D point clouds. The artifact injection idea is a heuristic and it does not really fix this shift. Reviewer NKTF also said the Anchor ControlNet claim is overstated, since it looks like a standard ControlNet with a visibility mask and not a real architectural change. The rebuttal adds some results, but it does not give a clear, principled answer to these issues, so there is no basis for acceptance. We encourage the authors to take into account the reviewers' comments and to resubmit to another venue.

**Reviewer Concerns:**

Addressed:

Baselines: The authors added comparisons with FloVD, Gen3C, and ReCamMaster, and the numbers look competitive on standard metrics.

Generalization: They also show EPiC can run on Wan2.1, so it is not tied to CogVideoX.

Outstanding:

Train-Inference Shift: The train inference gap remains. Training depends on clean 2D masks while inference uses noisy 3D points, and artifact injection still feels like a patch.

Overclaimed Novelty: The novelty claim around Anchor ControlNet is still not convincing as written, since it reads as ControlNet plus masking.

**Reviewer Scores:**

cqiR: Likely stays at 6.

yZGq: Likely stays at 4 since the train-inference gap is not resolved.

uFzJ: Likely stays at 6.

NKTF: Likely rises from 2 to 3. Added baselines help, but the novelty concern remains.

---

### Decision · Program_Chairs · 2026-01-26

Reject